# WHAT LAYERS WHEN: LEARNING TO SKIP COMPUTE IN LLMS WITH RESIDUAL GATES

**Filipe Laitenberger**[1,2], **Dawid Kopiczko**[3], **Cees G.M. Snoek**[2], **Yuki M. Asano**[3]

[1]Humboldt University Berlin
[2]Qualcomm-UvA Lab, University of Amsterdam
[3]FunAI Lab, University of Technology Nuremberg

## ABSTRACT

We introduce GateSkip, a simple residual-stream gating mechanism that enables token-wise layer skipping in decoder-only LMs. Each Attention/MLP branch is equipped with a sigmoid-linear gate that condense the branch's output before it re-enters the residual stream. During inference we rank tokens by the gate values and skip low-importance ones using a per-layer budget. While early-exit or router-based Mixture-of-Depths models are known to be unstable and need extensive retraining, our smooth, differentiable gates fine-tune stably on top of pretrained models. On long-form reasoning, we save up to 15% compute while retaining >90% of baseline accuracy. On instruction-tuned models we see accuracy gains at full compute and match baseline quality near 50% savings. The learned gates give insight into transformer information flow (e.g., BOS tokens act as anchors), and the method combines easily with quantization, pruning, and self-speculative decoding.

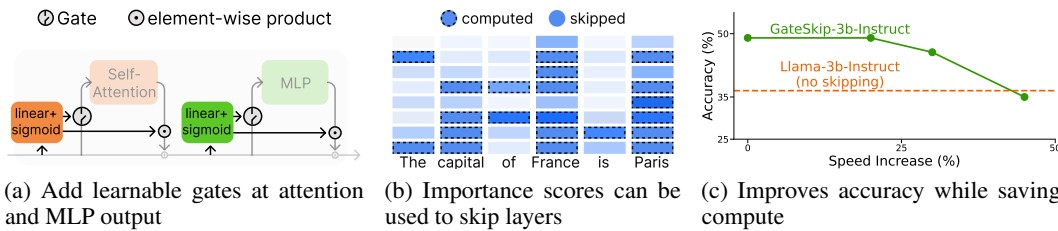

(a) Add learnable gates at attention and MLP output

(b) Importance scores can be used to skip layers

(c) Improves accuracy while saving compute

Figure 1: We introduce gating mechanisms that regulate the flow of information into the residual stream and can be used to skip layers altogether. **Our mechanism enhances downstream accuracy of instruction-tuned models even when skipping ∼25% of the model**.

## 1 INTRODUCTION

Large language models have transformed natural language processing, yet their rapid growth has created major challenges for efficient deployment. Current models allocate the same amount of computation to every token at every layer, regardless of difficulty. This uniform allocation is wasteful and makes it hard to deploy models in latency-sensitive or resource-limited environments. Adaptive compute aims to address this by using more resources where they matter and less where they do not.

Most prior approaches fall into two categories. Router-based methods, such as Mixture-of-Depths (Raposo et al., 2024), introduce specialized routing components that decide which layers to run. These rely on hard, discrete decisions that are often unstable and require careful balancing losses (Zoph et al., 2022). Early-exit methods attach auxiliary language modeling heads at intermediate layers and stop once a confidence threshold is reached (Schuster et al., 2022). These approaches alter pretrained hidden states, complicate training, and often fail to calibrate well (Bajpai & Hanawal, 2024a). Both approaches usually require implementation during pre-training.

We introduce **GateSkip**, a lightweight residual stream gating mechanism for decoder-only transformers. Each attention and MLP branch is equipped with a small linear gate and sigmoid activation that squashes the branch output before it is added back to the residual stream. During training, gates are optimized to remain sparse while preserving language modeling accuracy. At inference, token-level importance scores derived from the gates allow us to retain only the top tokens per layer using a quantile threshold, while skipped tokens copy their hidden states and key–value cache entries upward.

This design has several advantages. Because the gates are smooth and differentiable, GateSkip can be trained directly on top of pretrained models without destabilizing optimization. Since it operates entirely within the residual stream, it minimally perturbs existing representations. Moreover, the mechanism provides fine-grained control at both the token and module level, enabling nuanced allocation of compute. Finally, GateSkip is fully compatible with orthogonal efficiency techniques such as quantization, pruning, and self-speculative decoding.

We evaluate GateSkip on Llama 3.1 up to 8B parameters and Gemma 2 2B models across generative reasoning and log-likelihood benchmarks. On long-form reasoning tasks, GateSkip reduces computation by up to 15% while retaining more than 90% of the original accuracy. On instruction-tuned models, it improves accuracy at full compute and sustains this improvement under reduced budgets, matching baseline quality even with roughly 50% compute savings. Analysis of the learned gate values further reveals consistent patterns: early layers allocate more computation to the beginning-of-sequence token and punctuation, while deeper layers become increasingly selective and focus on content-bearing words.

Our contributions can be summarized as follows:

1. We propose GateSkip, a residual gating mechanism that enables token-wise layer skipping with smooth training and deterministic hard decisions at inference.

2. We demonstrate state-of-the-art compute–accuracy trade-offs on generative reasoning tasks, where prior adaptive compute methods often collapse.

3. We show that GateSkip composes seamlessly with quantization, pruning, and self-speculative decoding.

4. We provide an analysis of gate activations that sheds light on information flow within transformers.

GateSkip turns the residual stream into a simple yet effective control mechanism for adaptive depth, delivering efficiency gains without sacrificing stability or performance.

## 2 RELATED WORK

A growing body of work has addressed the inefficiency of ever-larger decoder-only Transformers by dynamically adapting computation on a per-token or per-sequence basis. Early efforts in **layer skipping** repurpose sparse routing from Mixture-of-Experts (MoE) to decide whether to execute each layer at all, yielding substantial compute savings without restarting training from scratch. Mixture-of-Depths (MoD) injects a router at every transformer layer to skip unimportant layers for each token (Raposo et al., 2024), while follow-up methods introduce soft token budgets (Zeng et al., 2023), sequence-level skipping (Wang et al., 2023), and frozen-backbone router fine-tuning (He et al., 2024). However, discrete routers can be unstable (Zoph et al., 2022; Fedus et al., 2022; Puigcerver et al., 2024; Panda et al., 2025), and most require training from scratch. In contrast to hard top-k routers with load-balancing losses, GateSkip uses fully differentiable residual gating, avoiding discrete routing during training while still yielding hard skips at test time.

In contrast, **early exiting** methods terminate inference once intermediate representations are deemed confident enough. Pioneered in encoder-only BERT models via entropy or agreement thresholds (Xin et al., 2020; Zhang et al., 2022; Zhou et al., 2020; Liu et al., 2020), this concept has been extended to encoder–decoder and decoder-only settings by supervising every layer with an auxiliary language modeling objective and exiting based on heuristic confidence measures (Tang et al., 2023; Schuster et al., 2022; Elbayad et al., 2020; Liu et al., 2021; Bae et al., 2023; Elhoushi et al., 2024; Del Corro et al., 2023). Yet these approaches fundamentally alter pretrained hidden representations through connector modules or self-distillation (Bajpai & Hanawal, 2024a;b; Ji et al., 2023). Where early exit

supervises intermediate LM heads and exits on confidence thresholds, GateSkip avoids auxiliary heads and maintains pretrained hidden spaces by training gates to compress post-module outputs.

Other efficiency techniques include **layer pruning** (e.g., ShortGPT removes redundant transformer blocks (Men et al., 2025)), **KV cache and token pruning** (e.g., query-driven pruning (Xu et al., 2025), token–precision trade-offs (Zhang et al., 2024)), and **quantization** (e.g., AWQ (Lin et al., 2023), SpQR (Dettmers et al., 2024), BiLLM (Huang et al., 2024)). These methods operate on different axes of efficiency and are thus orthogonal to our depth-adaptive approach. We further show that GateSkip is compatible with pruning as well as quantization, demonstrating its ability to combine with such methods.

## 3 GATESKIP

We propose adding a gating mechanism to the residual stream of decoder-only Transformer models and training it with an additional sparsity loss so that the gates learn to assess the importance of a certain Attention or MLP module given its preceding hidden state, as shown in Figure 1.

### 3.1 RESIDUAL GATING MECHANISM

The residual stream at layer $\ell$ in a transformer model can be described as the output $o_\ell \in \mathbb{R}^{B \times S \times H}$ of an Attention or MLP layer added to the hidden states $h_\ell \in \mathbb{R}^{B \times S \times H}$, resulting in the layer output $h_{\ell+1}$, with $B$ being the batch size, $S$ the sequence length, and $H$ the hidden dimension:

$$h_{\ell+1} = h_\ell + o_\ell \tag{1}$$

We propose supplementing the language model with a trainable gate $g$ which is a sigmoid-activated linear projection of the hidden states $h_\ell$:

$$h_{\ell+1} = h_\ell + o_\ell \odot g_\ell(h_\ell), \quad g_t(h_\ell) = \sigma(W_G h_\ell + b) \tag{2}$$

where $W_G \in \mathbb{R}^{H \times H}$ and $b \in \mathbb{R}^H$, $\sigma$ refers to the sigmoid function, and $g_\ell$ refers to the gate at layer $\ell$ which could theoretically be a shared gate across layers (cf. ablation experiments in §4.4) or separate gates for each layer. The gate is placed at the exit point of the module to the residual stream, after the output projection, making it perfectly compatible with multi-head attention or any variant thereof.

### 3.2 TRAINING OBJECTIVE

Training minimises a standard language–model loss (cross-entropy for next-token prediction)

$$\mathcal{L}_{\text{CE}} = -\frac{1}{|B|} \sum_{(x,y) \in B} \log p_\theta(y \mid x) \tag{3}$$

plus an explicit *gate-sparsity* penalty (L2 distance on gate activations)

$$\mathcal{L}_{\text{S}} = \frac{1}{N_L H} \sum_{\ell=1}^{N_L} \sum_{k=1}^{H} \big\| g_\ell(h_\ell)_k \big\|_2 \tag{4}$$

so that the overall loss becomes

$$\mathcal{L} = \mathcal{L}_{\text{CE}} + \lambda_S \, \mathcal{L}_{\text{S}}. \tag{5}$$

Here $N_L$ is the number of layers, $H$ the hidden dimension, and $\lambda_S$ balances accuracy and efficiency. Term (2) encourages each sigmoid gate $g_\ell(h_\ell)$ to stay close to zero, effectively **compressing the module output before it is re-added to the residual stream**. Backbone parameters $\theta$ and gate parameters are updated jointly with AdamW, with all weights being trainable.

## 3.3 Token selection

At step $t$ we allot a *fractional* budget $b_t \in (0, 1]$, the share of the $L$ tokens that may be *processed* in the current layer. For every token we collapse the current batch's gate vectors to scalar importance scores $\bar{g}_{\ell,i} = \frac{1}{H} \sum_k^H g_\ell(h_\ell)_{i,k}$ and form their empirical cumulative distribution function (CDF). We then compute the *threshold*,

$$\tau_t = \texttt{Quantile}\big(\{\bar{g}_{\ell,i}\}_{i=1}^L,\ 1 - b_t\big) \tag{6}$$

using linear interpolation between adjacent order-statistics (see Algorithm 3); thus the expected fraction of scores below $\tau_t$ equals the desired skip ratio $1 - b_t$. Tokens with $\bar{g}_{\ell,i} \leq \tau_t$ are **skipped**, copying the hidden state upwards $h_{\ell+1,i} = h_{\ell,i}$, and the rest is processed normally.

During **training** (see Algorithm 1) the budget decays linearly, as $b_t = b_1 - (b_1 - b_2)\frac{t}{T_{\text{total}}}$, so that the model learns to tolerate skipped hidden states. During **inference** (see Algorithm 2) we fix a single budget $\hat{b}$ once, re-use the same post-module gate scores for ranking, and apply the Top-$k$ only to tokens that have not yet emitted the end-of-sequence symbol. Additionally, when a token skips a layer, we upwards copy the KV cache items from the layer below in order to facilitate KV-cache reuse.

## 3.4 Implementation Details

We initialize the gates to ensure the model initially closely resembles the original pre-trained model. Specifically, we initialize the weights of the linear matrix $W_G$ around 0 using a Gaussian distribution with low standard deviation $\sigma = 0.01$, and set the biases $b$ to 5, so that the module's output remains approximately unchanged (as $\sigma(5) \approx 1$).

A key design decision is to place the gate at the module input for skipping decisions, but to train it on the module's output, i.e. we multiply the gate element-wise with the module output so that it receives its gradient signal downstream of the module. While the input to the gate is the same in both cases (the hidden states $h_\ell$), the learning signal differs compared to if the gate was placed at the entry point to the module. The gate is trained to incorporate minimal information from the module's output into the residual stream while maintaining language modeling performance, rather than determining which information should enter a module. We empirically found that training the gate after the module leads to better downstream performance (see §4).

Our method is numerically stable compared to other techniques based on hard binary routing (such as Mixture-of-Depths) (Zoph et al., 2022; Fedus et al., 2022; Puigcerver et al., 2024; Panda et al., 2025), providing effective control of information flow without introducing training instabilities or convergence issues (see §4 for experimental results).

The added gates introduce negligible parameter overhead to the model, e.g. 0.004% for separate gates with scalar output, and 4% for separate gates with hidden-state sized output on Llama-3.2-1b. Table 13 in Appendix H shows parameter overhead for each of the variants of GateSkip.

## 4 Experiments

We evaluate GateSkip on Llama-3 (Meta-AI, 2024) models of varying size as well as on Gemma 2 (Gemma-Team, 2024). We then perform ablation studies to isolate the impact of each component, compare against state-of-the-art layer-skipping and early-exit methods, and demonstrate compatibility with 4-bit quantization, self-speculative decoding, and structured pruning.

## 4.1 Experimental Setup

**Models and Training.** We primarily evaluate our method on Llama-3.2-1b, while also experimenting with Llama-3.2-3b, Llama-3.1-8b, and Gemma-2-2b to assess scalability and architecture independence. For all experiments, we fine-tune the pretrained backbone to train the gates while simultaneously adapting the model to task templates for easier downstream answer extraction. We set the sparsity loss weight $\lambda = 0.1$ and decay the token budget from 100% to 80% during training.

Table 1: Averaged results for loglikelihood-based and longer generation benchmarks for a random skipping baseline, prior adaptive compute methods and GateSkip on Llama-3.2-1b.

| saved compute | Generative Benchmarks | | | | | | Log-Likelihood Benchmarks | | | | |
|---|---|---|---|---|---|---|---|---|---|---|---|
| | 0% | 5% | 10% | 15% | 20% | 25% | 0% | 15% | 30% | 45% | 60% |
| Llama-1b | $30.97_{\pm1.44}$ | - | - | - | - | - | $49.12_{\pm0.05}$ | - | - | - | - |
| Llama-1b (random skipping) | - | $10.30_{\pm0.54}$ | $2.20_{\pm0.36}$ | $1.67_{\pm0.25}$ | $1.23_{\pm0.26}$ | $0.67_{\pm0.12}$ | - | $25.58_{\pm0.25}$ | $23.62_{\pm0.27}$ | $23.36_{\pm0.20}$ | $23.65_{\pm0.05}$ |
| CALM (hidden state saturation) | $3.43_{\pm0.49}$ | $3.43_{\pm0.49}$ | $3.43_{\pm0.49}$ | $3.43_{\pm0.49}$ | $3.43_{\pm0.49}$ | $3.43_{\pm0.49}$ | $30.73_{\pm0.00}$ | $30.73_{\pm0.00}$ | $30.73_{\pm0.00}$ | $30.73_{\pm0.00}$ | $30.73_{\pm0.00}$ |
| CALM (softmax) | $3.15_{\pm0.35}$ | $3.15_{\pm0.35}$ | $3.15_{\pm0.35}$ | $3.15_{\pm0.35}$ | $3.15_{\pm0.35}$ | $3.15_{\pm0.35}$ | $30.73_{\pm0.00}$ | $30.73_{\pm0.00}$ | $30.73_{\pm0.00}$ | $30.73_{\pm0.00}$ | $30.73_{\pm0.00}$ |
| FREE (hidden state saturation) | $11.57_{\pm0.63}$ | $11.57_{\pm0.63}$ | $11.57_{\pm0.63}$ | $11.57_{\pm0.63}$ | $11.57_{\pm0.63}$ | $11.57_{\pm0.63}$ | $36.02_{\pm0.01}$ | $36.02_{\pm0.01}$ | $36.02_{\pm0.01}$ | $36.02_{\pm0.01}$ | $36.02_{\pm0.01}$ |
| FREE (softmax) | $10.70_{\pm0.00}$ | $10.70_{\pm0.00}$ | $10.70_{\pm0.00}$ | $10.70_{\pm0.00}$ | $10.70_{\pm0.00}$ | $10.70_{\pm0.00}$ | $36.02_{\pm0.01}$ | $36.02_{\pm0.01}$ | $36.02_{\pm0.01}$ | $36.02_{\pm0.01}$ | $36.02_{\pm0.01}$ |
| LayerSkip | $10.65_{\pm1.55}$ | $10.65_{\pm1.55}$ | $10.65_{\pm1.55}$ | $10.65_{\pm1.55}$ | $10.65_{\pm1.55}$ | $10.65_{\pm1.55}$ | $38.25_{\pm0.02}$ | $38.25_{\pm0.02}$ | $\mathbf{38.25_{\pm0.02}}$ | $\mathbf{38.25_{\pm0.02}}$ | $\mathbf{38.25_{\pm0.02}}$ |
| MoD (router-tuned) | $0.00_{\pm0.00}$ | $0.00_{\pm0.00}$ | $0.00_{\pm0.00}$ | $0.00_{\pm0.00}$ | $0.00_{\pm0.00}$ | $0.00_{\pm0.00}$ | $\mathbf{51.34_{\pm0.05}}$ | $23.68_{\pm0.03}$ | $22.98_{\pm0.23}$ | $22.94_{\pm0.09}$ | $22.86_{\pm0.04}$ |
| MoD | $20.83_{\pm5.22}$ | $9.45_{\pm4.63}$ | $4.90_{\pm2.92}$ | $3.96_{\pm1.67}$ | $3.37_{\pm1.33}$ | $2.91_{\pm1.33}$ | $44.18_{\pm0.66}$ | $31.88_{\pm1.48}$ | $29.33_{\pm2.42}$ | $26.72_{\pm0.39}$ | $26.00_{\pm0.58}$ |
| SkipLayer | $0.70_{\pm0.60}$ | $0.24_{\pm0.17}$ | $0.04_{\pm0.00}$ | $0.05_{\pm0.05}$ | $0.05_{\pm0.05}$ | $0.04_{\pm0.04}$ | $32.00_{\pm1.46}$ | $23.03_{\pm0.12}$ | $23.60_{\pm0.44}$ | $23.67_{\pm0.47}$ | $23.41_{\pm0.09}$ |
| GateSkip (ours) | $23.53_{\pm1.28}$ | $\mathbf{23.05_{\pm1.24}}$ | $\mathbf{22.57_{\pm1.24}}$ | $\mathbf{22.14_{\pm1.36}}$ | $\mathbf{20.37_{\pm1.18}}$ | $\mathbf{17.67_{\pm0.74}}$ | $47.35_{\pm0.22}$ | $\mathbf{38.86_{\pm0.45}}$ | $31.74_{\pm2.04}$ | $27.93_{\pm0.47}$ | $26.49_{\pm0.57}$ |

Table 2: Accuracy at 15% saved compute for log-likelihood-based and generative benchmarks for a random skipping baseline, prior adaptive compute methods and GateSkip on Llama-3.2-1b.

| | CSQA (Gen.) | GSM8K (Gen.) | MMLU Stem | HellaSwag | CSQA | PIQA | Open-BookQA | Wino-Grande |
|---|---|---|---|---|---|---|---|---|
| Llama-1b (random skipping) | $3.27_{\pm0.53}$ | $0.07_{\pm0.09}$ | $24.00_{\pm0.26}$ | $30.87_{\pm0.57}$ | $20.27_{\pm1.09}$ | $53.67_{\pm0.94}$ | $15.74_{\pm0.66}$ | $51.13_{\pm0.34}$ |
| CALM (hidden state saturation) | $5.40_{\pm1.23}$ | $1.47_{\pm0.25}$ | $21.85_{\pm0.00}$ | $34.00_{\pm0.00}$ | $19.00_{\pm0.00}$ | $61.60_{\pm0.00}$ | $16.20_{\pm0.00}$ | $50.60_{\pm0.00}$ |
| CALM (softmax) | $4.70_{\pm0.90}$ | $1.60_{\pm0.20}$ | $21.85_{\pm0.00}$ | $34.00_{\pm0.00}$ | $19.00_{\pm0.00}$ | $61.60_{\pm0.00}$ | $16.20_{\pm0.00}$ | $50.60_{\pm0.00}$ |
| FREE (hidden state saturation) | $17.07_{\pm0.57}$ | $6.07_{\pm0.75}$ | $21.26_{\pm0.01}$ | $41.33_{\pm0.09}$ | $19.40_{\pm0.00}$ | $70.93_{\pm0.09}$ | $21.87_{\pm0.09}$ | $50.47_{\pm0.09}$ |
| FREE (softmax) | $16.40_{\pm0.00}$ | $5.00_{\pm0.00}$ | $21.27_{\pm0.02}$ | $41.30_{\pm0.10}$ | $19.40_{\pm0.00}$ | $71.00_{\pm0.00}$ | $21.90_{\pm0.10}$ | $50.30_{\pm0.10}$ |
| LayerSkip | $16.40_{\pm1.80}$ | $4.90_{\pm1.30}$ | $21.31_{\pm0.00}$ | $\mathbf{42.40_{\pm0.00}}$ | $19.00_{\pm0.00}$ | $\mathbf{72.33_{\pm0.09}}$ | $20.80_{\pm0.00}$ | $\mathbf{56.33_{\pm0.09}}$ |
| MoD (router-tuned) | $0.00_{\pm0.00}$ | $0.00_{\pm0.00}$ | $23.09_{\pm0.21}$ | $28.87_{\pm0.35}$ | $19.92_{\pm0.24}$ | $53.60_{\pm0.41}$ | $13.81_{\pm0.54}$ | $49.08_{\pm1.36}$ |
| MoD | $6.14_{\pm3.06}$ | $1.77_{\pm0.29}$ | $26.00_{\pm0.66}$ | $38.21_{\pm1.55}$ | $20.86_{\pm0.94}$ | $59.88_{\pm2.21}$ | $18.58_{\pm1.36}$ | $52.15_{\pm0.73}$ |
| SkipLayer | $0.10_{\pm0.10}$ | $0.00_{\pm0.00}$ | $21.82_{\pm0.42}$ | $27.76_{\pm0.80}$ | $19.06_{\pm1.67}$ | $50.51_{\pm0.50}$ | $16.40_{\pm1.23}$ | $48.72_{\pm1.11}$ |
| GateSkip (ours) | $\mathbf{35.25_{\pm1.81}}$ | $\mathbf{9.03_{\pm1.31}}$ | $\mathbf{30.86_{\pm1.18}}$ | $39.05_{\pm1.25}$ | $\mathbf{36.16_{\pm1.23}}$ | $70.45_{\pm0.60}$ | $\mathbf{22.69_{\pm0.43}}$ | $52.66_{\pm0.82}$ |

Training employs the AdamW optimizer (Loshchilov & Hutter, 2017). A full list of hyperparameters can be found in Appendix K. All libraries and their respective versions used for our experiments are listed in Appendix L. Instructions for code access can be found in Appendix J.

**Generative benchmarks.** We fine-tune on the train sets of CommonsenseQA (Talmor et al., 2019) and GSM8K (Cobbe et al., 2021) questions with chain-of-thought traces generated by Nemotron-70B (Anonymous, 2024; Reasoning, 2024; Wang et al., 2024), masking the loss on the question portion so that the model learns both reasoning and answer extraction. For evaluation we measure zero-shot accuracy on the GSM8K and CommonsenseQA test sets using the same prompt template. We sweep the inference budget $\hat{b}$ over $\{1.00, 0.95, 0.90, 0.85, 0.80, 0.75\}$, corresponding to compute-savings of $\{0\%, 5\%, 10\%, 15\%, 20\%, 25\%\}$; since the realized savings sometimes fall between these targets, we linearly interpolate the measured accuracies to report performance at the exact percentages listed above.

**Log-likelihood benchmarks.** We fine-tune on FineWeb data (Penedo et al., 2024) with the same hyperparameters as above. For evaluation we measure five-shot log-likelihood accuracy on MMLU (Hendrycks et al., 2021), HellaSwag (Zellers et al., 2019), CommonsenseQA (Talmor et al., 2019), PIQA (Bisk et al., 2019), OpenBookQA (Mihaylov et al., 2018), and Wino-Grande (Sakaguchi et al., 2021) using LM Evaluation Harness (Gao et al., 2024). We sweep $\hat{b}$ over $\{1.00, 0.85, 0.70, 0.55, 0.40\}$ for compute-savings of $\{0\%, 15\%, 30\%, 45\%, 60\%\}$, again using linear interpolation to report exact savings levels.

**Variance estimate.** We perform five seeds per configuration, measuring mean and standard deviation on each metric. All baselines (random skipping, MoD router-tuning, CALM variants) are trained and evaluated with the identical data splits, hyperparameters, inference budgets, and interpolation procedure described above, ensuring a fair comparison.

## 4.2 COMPARISON TO BASELINE

We begin by evaluating GateSkip against a straightforward token-level heuristic: *random skipping*. At each layer, a fixed fraction of tokens is selected uniformly at random to be omitted from further computation. All experiments use the Llama-3.2-1b backbone.

Table 3: GateSkip on Llama Instruct.

| | Generative Benchmarks | | | | Log-Likelihood Benchmarks | | | |
|---|---|---|---|---|---|---|---|---|
| saved compute | 0% | 20% | 30% | 45% | 0% | 15% | 30% | 60% |
| Llama-3b-Instruct | 36.5 | - | - | - | **46.3** | - | - | - |
| Llama-3b-Instruct + random skipping | - | 0.5 | 0.1 | 0.1 | - | 34.7 | 30.4 | 30.4 |
| GateSkip (Llama-3b-Instruct) | **49.0** | **49.0** | **45.6** | **35.0** | 36.7 | **38.8** | **32.9** | **31.0** |

Table 4: GateSkip on out-of-domain generative tasks.

| | MMLU-Gen | | | | | | PIQA-Gen | | | | |
|---|---|---|---|---|---|---|---|---|---|---|---|
| saved compute | 0% | 10% | 15% | 20% | 30% | 45% | 0% | 10% | 20% | 25% | 30% |
| Llama-1b | **22.8** | - | - | - | - | - | **22.9** | - | - | - | - |
| Llama-1b (random skipping) | - | 7.5 | 2.0 | 1.0 | 0.3 | 0.1 | - | 7.0 | 0.8 | 1.2 | 0.3 |
| GateSkip | 14.0 | **15.6** | **18.6** | **15.9** | **12.8** | **5.1** | 14.3 | **16.9** | **21.8** | **29.8** | **29.5** |

Table 1 presents averaged accuracies across multiple compute-savings levels, while Table 2 reports performance at exactly 15% saved compute. Random skipping collapses generative accuracy to under 10% even at modest budgets (5–10% savings), while GateSkip retains the majority of performance. Additional results on LAMBADA are provided in Appendix A, showing that GateSkip maintains stable perplexity and accuracy under compute constraints, while random skipping collapses sharply. Moreover, results on translation are presented in Appendix B, showing that GateSkip exhibits analogous performance improvements compared to baseline performance.

## 4.3 COMPARISON TO PRIOR STATE-OF-THE-ART

Having established the superiority of GateSkip over naive heuristics, we now compare it against prior adaptive-depth methods under identical fine-tuning and evaluation settings: (1) **Mixture-of-Depths (MoD)** with Router Tuning, (2) **CALM** in its hidden state saturation and softmax variants, (3) **FREE** in both variants, and (4) static skipping approaches such as **LayerSkip** and **SkipLayer**. Results averaged across different benchmarks as well as task-level accuracy at exactly 15% saved compute can be seen in Tables 1 and 2 respectively.

On generative benchmarks, GateSkip achieves 36.7% accuracy on CSQA and 9.7% on GSM8K—over four times higher than MoD and orders of magnitude above CALM. FREE maintains strong log-likelihood scores across all budgets but remains flat on generation (11.9), while LayerSkip and SkipLayer collapse quickly on longer reasoning, with accuracies of only 13.1 and $\leq 2.2$ respectively. In contrast, GateSkip sustains high generative accuracy while remaining competitive on log-likelihood evaluations.

Regarding log-likelihood benchmarks, across all metrics GateSkip either matches or outperforms MoD, CALM, and the static baselines, while FREE achieves similar log-likelihood scores but without corresponding generative performance. These results confirm that GateSkip consistently delivers a superior compute–accuracy trade-off across both reasoning and multiple-choice tasks. On instruction-tuned models (Table 3), GateSkip improves generative accuracy over the Llama-3b-Instruct baseline even under aggressive budgets (e.g., +12.5 points at 0–20% saved compute) while also matching or slightly improving log-likelihood accuracy at 15–60% savings.

We tested on log-likelihood-based benchmarks following prior literature. However, real-world scenarios would demand robustness to longer generation which is why we performed such experiments as well. Notably, there is a visible discrepancy between log-likelihood and generative tasks for prior methods, whereas GateSkip retains accuracy significantly better over longer generation.

Beyond the standard suites, out-of-domain generative evaluations (Table 4) show GateSkip retains competitive performance on MMLU-Gen at reduced compute and, notably, exceeds the unadapted baseline on PIQA-Gen at 20–30% saved compute (29.8–29.5 vs. 22.9). This suggests that targeted

Table 5: Ablation of GateSkip's design choices on Llama-3.2-1b.

| Compute saved | Generative Benchmarks | | | | | | Log-Likelihood Benchmarks | | | | |
|---|---|---|---|---|---|---|---|---|---|---|---|
| | 0% | 5% | 10% | 15% | 20% | 25% | 0% | 15% | 30% | 45% | 60% |
| **Gate output shape** | | | | | | | | | | | |
| Scalar gates | 23.6 | 22.7 | 21.7 | 20.4 | 15.9 | 14.2 | 42.8 | 36.8 | **33.7** | **30.9** | **31.8** |
| Vector gates (default) | **26.8** | **25.5** | **24.2** | **23.2** | **23.6** | **19.8** | **44.3** | **37.8** | 32.7 | 30.8 | 31.2 |
| **Gate parameter sharing** | | | | | | | | | | | |
| Shared | 21.8 | 21.4 | 21.0 | 20.7 | 19.5 | 15.5 | **44.3** | **38.4** | **35.4** | **31.8** | **31.7** |
| Separate (default) | **26.8** | **25.5** | **24.2** | **23.2** | **23.6** | **19.8** | **44.3** | 37.8 | 32.7 | 30.8 | 31.2 |
| **Skipping strategy** | | | | | | | | | | | |
| Only attention layers | 26.8 | 23.1 | 19.2 | 14.9 | 10.8 | 6.8 | 44.3 | 37.5 | 30.8 | 30.6 | 30.3 |
| Only MLP layers | 26.8 | 24.1 | 18.6 | 7.8 | 1.2 | 0.4 | 44.3 | 32.0 | 30.4 | **32.1** | **32.0** |
| Skip entire layer (attn gate) | 26.8 | **25.5** | **24.2** | **23.2** | **23.6** | **19.8** | 44.3 | **37.8** | 32.7 | 30.8 | 31.2 |
| Every-second layer | 26.8 | **25.5** | 22.8 | 15.0 | 10.1 | 9.5 | 44.3 | 35.5 | **33.3** | 31.6 | 31.9 |
| Skip all layers (default) | 26.8 | **25.5** | **24.2** | **23.2** | **23.6** | **19.8** | 44.3 | **37.8** | 32.7 | 30.8 | 31.2 |
| **Gate architecture** | | | | | | | | | | | |
| MLP-based gate | 24.3 | 24.3 | **24.4** | 18.5 | 17.0 | 15.6 | 44.0 | 33.9 | 31.8 | 30.3 | 30.7 |
| Linear-sigmoid gate (default) | **26.8** | **25.5** | 24.2 | **23.2** | **23.6** | **19.8** | **44.3** | **37.8** | **32.7** | **30.8** | **31.2** |
| **Gate placement** | | | | | | | | | | | |
| Gate before module (entry) | 21.1 | 5.5 | 1.3 | 1.0 | 0.9 | 0.8 | 40.0 | 35.7 | **33.2** | **31.6** | 30.8 |
| Gate after module (default) | **26.8** | **25.5** | **24.2** | **23.2** | **23.6** | **19.8** | **44.3** | **37.8** | 32.7 | 30.8 | **31.2** |
| **Gate Loss** | | | | | | | | | | | |
| L1 | 23.0 | 22.6 | 22.2 | 21.4 | 18.7 | 17.0 | **47.5** | 37.6 | 29.2 | 27.1 | 26.3 |
| KL-div. | **36.0** | 18.1 | 11.9 | 9.8 | 7.9 | 5.3 | 45.6 | 26.6 | 24.5 | 23.0 | 23.0 |
| L2 (default) | 26.8 | **25.5** | **24.2** | **23.2** | **23.6** | **19.8** | 44.3 | **37.8** | **32.7** | **30.8** | **31.2** |
| **Frozen Backbone** | | | | | | | | | | | |
| Frozen B. | 14.9 | 14.1 | 13.3 | 12.7 | 12.5 | 12.6 | **45.6** | 37.5 | 26.7 | 24.0 | 23.9 |
| Unfrozen B. (default) | **26.8** | **25.5** | **24.2** | **23.2** | **23.6** | **19.8** | 44.3 | **37.8** | **32.7** | **30.8** | **31.2** |

token-level allocation can translate into quality gains on certain OOD generative tasks, not merely lossless efficiency.

## 4.4 COMPONENT ABLATIONS

To understand the contribution of each design choice in GateSkip, we perform a series of controlled ablations on Llama-3.2-1b. Table 5 summarizes the impact of varying the gate parameterization, skipping granularity, gate architecture, and gate placement on both our generative and log-likelihood benchmarks at multiple compute-savings levels.

**General Oberservations.** Since we condense the output of modules, downstream performance changes even at no skipping. Hence, different modifications of our method will have differing effects on performance at 0% skipping.

**Gate Output Shape.** We compare two forms of gating output: (1) *Vector-gates*, which produce an $H$-dimensional output per residual branch, and (2) *Scalar-gates*, which produce a single gating value per branch. At 15% compute-savings on our generative benchmarks, vector-gates achieve 23.2% accuracy, compared to only 20.4% for scalar-gates, confirming that a full-dimensional gate yields more precise control.

**Gate Parameter Sharing.** Focusing on the vector-gate design, we then compare (1) *Per-layer vector-gates*: a distinct gate for each Attention and MLP module, versus (2) *Shared vector-gates*: one gate shared across all Attention modules and one across all MLP modules. Per-layer vector-gates again lead, with 23.2% at 15% savings, while shared vector-gates lag at 20.7%, showing the value of layer-specific parameters.

**Skipping Granularity.** Ablating which sub-modules can be skipped reveals that attention and MLP layers are both essential. When only attention layers are skipped, accuracy falls to 14.9% at a 15% compute reduction; skipping only MLP layers drops performance even further, to 7.8% under the same budget. Applying a single gate over the entire layer performs on par with our default per-module

Table 6: GateSkip on different model sizes and architectures.

| | Generative Benchmarks | | | | | | Log-Likelihood Benchmarks | | | | |
|---|---|---|---|---|---|---|---|---|---|---|---|
| saved compute | 0% | 5% | 10% | 15% | 20% | 25% | 0% | 15% | 30% | 45% | 60% |
| Llama-3.2-1B | 26.8 | 25.5 | 24.2 | 23.2 | 23.6 | 19.8 | 44.3 | 37.8 | 32.7 | 30.8 | 31.2 |
| Llama-3.2-3B | 45.0 | 44.4 | 43.9 | 43.3 | 42.7 | 42.1 | 55.9 | 35.6 | 31.4 | 29.3 | 30.4 |
| Llama-3.1-8B | 57.3 | 56.5 | 55.8 | 55.0 | 54.3 | 53.6 | 62.8 | 44.2 | 34.3 | 31.1 | 29.1 |
| Gemma-2-2B | 38.0 | 37.4 | 36.7 | 36.1 | 35.4 | 34.8 | 52.9 | 44.1 | 35.6 | 31.7 | 30.9 |

approach, but skipping every second layer leads to a steep decline, from 23.2% down to 15.0% at the 15% savings level. These results underscore the importance of fine-grained, per-module control.

**Gate Architecture.** We also tested a small MLP in place of our linear–sigmoid gate, but despite the extra parameters it underperforms. At 15% compute savings the MLP-based gate achieves only 18.5% on generative tasks compared to 23.2% with the linear gate, and 33.9% on log-likelihood benchmarks versus 37.8%. We therefore retain the simpler, more effective linear projection.

**Gate Placement.** Placing the gate before the module proved disastrous: at a 5% compute reduction entry-point gating yields just 5.5% accuracy compared to 25.5% when the gate is applied after the module. This dramatic gap confirms that post-module residual gating is crucial for stable, effective learning, as discussed in Section 3.4.

**Gate Loss Type.** Replacing our L2-loss on gate activations with an L1 loss mostly underperforms, whereas it achieves a higher loglikelihood accuracy at 0% compute savings. Furthermore, we tested a layer-wise KL divergence regularizer that matches the average gate activation per layer to a target budget, encouraging control of the overall fraction of retained tokens rather than only shrinking individual gates:

$$\mathcal{L}_{\text{layer-KL}} = \sum_{\ell} \text{KL}(\text{Bern}(\bar{g}_\ell) \parallel \text{Bern}(b_{\text{train}})), \text{ where } \bar{g}_\ell = \frac{1}{BSH} \sum_{b,i,k} g_\ell(h_\ell)_{b,i,k}.$$

While this loss clearly outperforms our L2 and L1 variants at no compute savings in the generative settings, it performs poorly at even low compute savings.

**Frozen Backbone.** To test the extent to which the backbone adapts, we froze a backbone finetuned on our dataset (for fair comparison) and trained gates on the frozen backbone. In this frozen setting, downstream accuracy substantially decreases. This gives evidence that backbone adaptation plays a major role in the skipping task: the backbone may by itself not reveal all information needed for skipping, necessitation adaptation so that it saves relevance cues in the hidden states which in turn condition the gates.

**GateSkip on varying model sizes and architectures.** To test scalability, we applied GateSkip to larger Llama models (3b and 8b) and observed consistent performance patterns (see Table 6). The results reveal that for larger architectures, the model is capable of skipping increasingly more tokens without decreasing performance. For instance, Llama-3.2-3B with GateSkip can save 37.3% computation while retaining 91.5% of its baseline GSM8K (Gen.) performance and 87.3% of its baseline CSQA (Gen.) performance. Moreover, comparisons between Llama and Gemma architectures reveal that the compute-accuracy trade-off generalizes across both model families. The instruction-tuned and LAMBADA results mirror these trends: larger or instruction-adapted backbones benefit more from budgeted token selection, maintaining strong generation quality where random or uniform skipping fails, and confirming that GateSkip's gains persist across usage styles (chat/instruction), lengths, and domains.

### 4.5 COMPATIBILITY WITH OTHER EFFICIENCY TECHNIQUES

We evaluate compatibility with various orthogonal efficiency techniques. Specifically, we show that GateSkip is compatible with 4-bit quantization, speculative decoding, and structured pruning.

**Compatibility with 4-bit Quantization.** To test compatibility with quantization, we apply 4-bit quantization to Llama-3.2-3b trained with GateSkip (Table 7) and downstream evaluate the quantized

model as before. The results demonstrate that GateSkip remains effective when combined with quantization, with performance curves closely tracking those of the 32-bit model. On generative benchmarks, the quantized model retains 94.4% of the original accuracy at 0% skipping ratio, 96.1% at 15%, and 97.3% at 25% skipping ratio. The performances for log-likelihood-based benchmarks exactly match.

Table 7: Quantization robustness (Llama-3.2-3 B).

| | Generative Benchmarks | | | | | | Log-Likelihood Benchmarks | | | | |
|---|---|---|---|---|---|---|---|---|---|---|---|
| saved compute | 0% | 5% | 10% | 15% | 20% | 25% | 0% | 15% | 30% | 45% | 60% |
| 32-bit + GateSkip | 45.0 | 44.4 | 43.9 | 43.3 | 42.7 | 42.1 | 55.9 | 35.6 | 31.4 | 29.3 | 30.4 |
| 4-bit + GateSkip | 42.5 | 42.2 | 41.9 | 41.6 | 41.3 | 41.0 | 55.9 | 35.6 | 31.4 | 29.3 | 30.4 |

**Compatibility with Speculative Decoding.** Adding speculative decoding boosts Log-Likelihood performance substantially at moderate savings: at 15% and 30% saved compute, it outperforms vanilla GateSkip (Table 8).

Table 8: GateSkip combined with self-speculative decoding compared to GateSkip alone and Layer-Skip. Metrics are log-likelihood accuracy (LL) at fixed saved-compute levels.

| | LL@15% | LL@30% | LL@45% | LL@60% |
|---|---|---|---|---|
| GateSkip | 37.8 | 32.7 | 30.8 | 31.2 |
| GateSkip + self-speculative decoding | **39.4** | **39.4** | 31.4 | 30.7 |

**Compatibility with Structured Pruning.** Structured pruning (Men et al., 2024) reduces absolute Log-Likelihood performance, but GateSkip remains notably stronger than the pruned backbone baseline at every budget (Table 9). At 0% savings, *GateSkip + pruning* trails unpruned GateSkip, as expected, yet still exceeds the *Default Llama1b + pruning* by +6.4 LL points.

Table 9: GateSkip with and without additional structured pruning of 25% of transformer blocks (ShortGPT).

| | LL@0% | LL@15% | LL@30% | LL@45% |
|---|---|---|---|---|
| GateSkip | **44.3** | **37.8** | **32.7** | **30.8** |
| GateSkip + ShortGPT pruning | 31.5 | 31.1 | 31.9 | **30.8** |
| Default Llama1b + ShortGPT pruning | 25.1 | 25.4 | 25.4 | 26.3 |

## 4.6 END-TO-END EFFICIENCY GAINS IN REAL-WORLD SCENARIOS

Table 10: End-to-end latency and throughput at different token skipping levels.

| % Tokens Skipped | 5% | 15% | 25% | 35% | 50% | 70% |
|---|---|---|---|---|---|---|
| Latency (s) | 607.29 | 606.01 | 559.81 | 571.79 | 521.68 | 449.85 |
| Throughput (tokens/s) | 2697.87 | 2703.57 | 2926.71 | 2865.39 | 3140.61 | 3642.11 |

Table 10 shows end-to-end latency and throughput measurements for our Llama-1b GateSkip model evaluated on GSM8K and CommonsenseQA-Gen using an optimized vLLM environment. The results show that GateSkips theoretical FLOP savings translate into analogous real-world efficiency gains. Analogously, Figure 7 in Appendix D shows consistently decreasing wall-clock time with lower FLOPs per token.

## 4.7 ANALYSIS OF GATE VALUES

Appendix C shows that GateSkip concentrates compute on BOS/punctuation anchors and salient content words, with deeper layers becoming increasingly selective. Gate scores exhibit tight, layer-specific distributions separated by tiny margins, motivating our per-layer quantile thresholds. The same scores also localize policy-relevant spans, suggesting value for interpretability and safety.

## 5 LIMITATIONS

Our study targets 1–8 B-parameter decoder-only LLMs and evaluates on English reasoning, translation, and general language modeling. We report theoretical FLOP reductions in the main text, as they more faithfully capture methodological differences and enable fair comparison across approaches, while also presenting end-to-end efficiency gains in § 4.6. An ethics statement and LLM usage is disclosed in Appendix E.

## 6 CONCLUSION

We introduced GateSkip, a residual gating mechanism that enables token-wise layer skipping in decoder-only transformers. GateSkip achieves up to 15–20% compute savings while retaining more than 90% accuracy on long-form reasoning, and on instruction-tuned models it improves accuracy at full compute and matches baseline quality with nearly 50% savings. These results establish a new state of the art in adaptive compute, particularly in generative settings where prior methods collapse.

Beyond efficiency, the learned gates provide insight into transformer information flow, consistently allocating more compute to BOS and punctuation tokens as well as salient content tokens.

GateSkip turns the residual stream into a stable and practical control mechanism for adaptive depth, offering real efficiency gains without destabilizing training or disrupting pretrained representations.

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

## A    ADDITIONAL RESULTS: LAMBADA

Table 11: Perplexity and accuracy on LAMBADA under different saved compute levels. GateSkip degrades gracefully while random skipping collapses.

| | Perplexity | | | | Accuracy | | | |
|---|---|---|---|---|---|---|---|---|
| saved compute | 0% | 10% | 20% | 30% | 0% | 10% | 20% | 30% |
| Llama-1b | 89.8 | - | - | - | 31.0 | - | - | - |
| Llama-1b (random skipping) | - | 2210.0 | 491000.0 | 10300000.0 | - | 12.97 | 2.36 | 0.12 |
| GateSkip | **23.7** | **70.9** | **588.0** | **9180.0** | **40.4** | **28.4** | **15.8** | **4.6** |

LAMBADA evaluates long-context language modeling where error accumulation typically amplifies weaknesses in adaptive methods. GateSkip substantially outperforms default Llama at 0% skipping, as well as random skipping across both perplexity and accuracy, demonstrating stable degradation under compute savings rather than catastrophic collapse. This supports our main claim that residual gating provides robustness on long generation tasks.

## B    ADDITIONAL RESULTS: GATESKIP ON TRANSLATION

Table 12: Translation – WMT16 English→Romanian. (a) Baseline BLEU at 0 % skipping. (b) Largest compute reduction that still preserves ≥ 90 % of that BLEU (higher is better).

| | WMT16-EN-RO | | | | | |
|---|---|---|---|---|---|---|
| saved compute | 0% | 5% | 10% | 15% | 20% | 25% |
| Llama-1b | **0.57** | - | - | - | - | - |
| Llama-1b (random skipping) | - | 0.36 | 0.14 | 0.03 | 0.01 | 0.01 |
| GateSkip | 0.51 | **0.43** | **0.37** | **0.32** | **0.28** | **0.2** |

To evaluate GateSkip on a sequence-to-sequence task, we fine-tune Llama-3.2-1b with GateSkip (separate vector gates at each layer) on the WMT16 English–Romanian training set for one hour, using the same hyperparameters as in our initial experiments. Table 12 shows BLEU scores on the WMT16 test set under varying compute-savings. Even with 10% and 15% of the layers skipped, GateSkip retains 65% and 63% of the full-compute BLEU (0.37/0.32 vs. 0.57), exhibiting a significantly more advantageous trade-off between efficiency and translation quality than the random skipping baseline.

## C    QUALITATIVE ANALYSIS OF GATE VALUES

The preceding chapters demonstrated that *GateSkip* can remove a double-digit fraction of Transformer FLOPs while maintaining competitive downstream accuracy. In this chapter, we turn from quantitative evaluation to qualitative analysis. Concretely, we analyze the distribution of learned gate values for individual sequences and ask what they reveal about (i) information flow within the residual stream, (ii) the model's implicit safety heuristics, and (iii) the practical utility of gate values as an interpretability signal (**RQ5**). Lastly we look into the overall distribution of gate values across entire datasets to gain insight about effective token budgeting.

### C.1    VISUALIZATION SET-UP

Unless stated otherwise we inspect a Llama-3.2-1B model fine-tuned with *shared vector gates*, one for each Attention and one for each MLP module. For each token $i$ and layer $\ell$ we compute the scalar importance

$$\bar{g}_{\ell,i} = \frac{1}{H} \sum_{k=1}^{H} g_{\ell,i,k}, \qquad (7)$$

where $H$ is the hidden dimension. We create heatmaps showing the average gate value for each token and layer, separated into Attention and MLP modules for better clarity of the patterns distinct to each module type. The resulting heatmaps are shown in Figures 2 and 5. Darker colors correspond to higher compute allocation.

## C.2 BOS Tokens and Punctuation as Structural Anchors

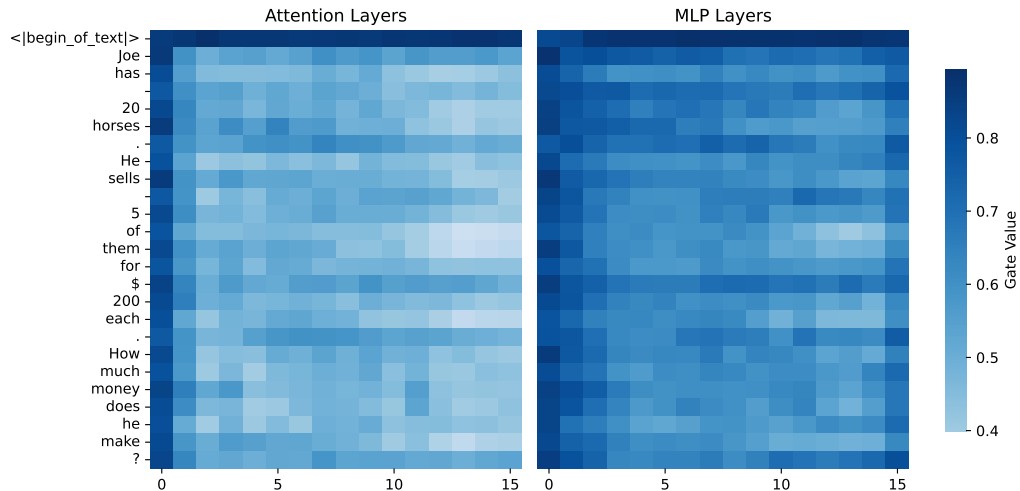

Figure 2: Mean gate value for each token in a sample sequence (Llama-3.2-1b, vector gate, shared across layers). The first Attention and MLP layer, as well as BOS tokens and punctuation, receive elevated importance. This hints at the model using BOS tokens to "dilute" attention, avoiding over-mixing. Another hypothesis is that punctuation and BOS tokens are used as critical reference points for establishing contextual boundaries.

Figure 2 shows mean gate values for our Llama-1b model for the sample sequence:

*Joe has 20 horses. He sells 5 of them for $200 each. How much money does he make?*

We split the importance scores into one sub-figure for the Attention and another for the MLP layers to highlight the patterns present. Moreover, we make several key observations:

1. Functional tokens (prepositions, pronouns, articles) receive consistently lower gate values than content words, particularly in later layers.

2. The first layer maintains high importance across all tokens while deeper layers become more selective.

3. Beginning-of-sequence (BOS) tokens and punctuation receive exceptionally high importance across all layers.

**Quantitative Analysis.** To quantitatively confirm BOS token prominence, we collect gate activations for all vocabulary items while evaluating our Llama-1b model with shared vector gates across the test sets of GSM8K and CommonsenseQA, as well as the PIQA's validation set. Moreover, we perform this test at varying skipping ratios, i.e. one run with 0% and one with 30% skipping. We average the collected activations across layers and samples to obtain a single number per vocabulary item. The sorted top-10 tokens with highest activations are shown in Figure 3 for 0% skipping (left) and 30% skipping (right). Across both skipping levels, the BOS token attains the highest gate activation, with a considerable margin ($\approx 0.01$) to the tokens that follow (subsequent margins lie below $<0.001$). In turn, the remaining top activations are rather uniform, with the only notable jump existing between the BOS token's and the subsequent token's activation. This quantitative test validates our earlier observation that BOS tokens receive elevated importance by our gates.

The consistently high BOS activations in our findings cast doubt on Guo et al. (2024)'s and Xiao et al. (2024)'s hypothesis that models allocate "excess" attention to non-meaningful tokens. If BOS tokens contained primarily redundant information, our gates would naturally assign them lower importance. While Clark et al. (2019) suggest BOS tokens accumulate sentence-level information, this cannot explain the initial BOS token's importance due to causal attention constraints.

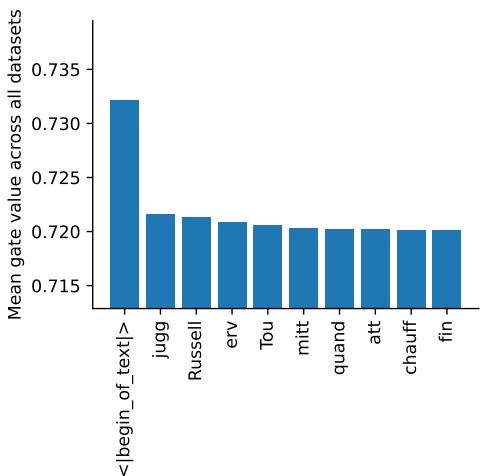
(a) Top-10 tokens with highest gate activation at 0% skipping ratio

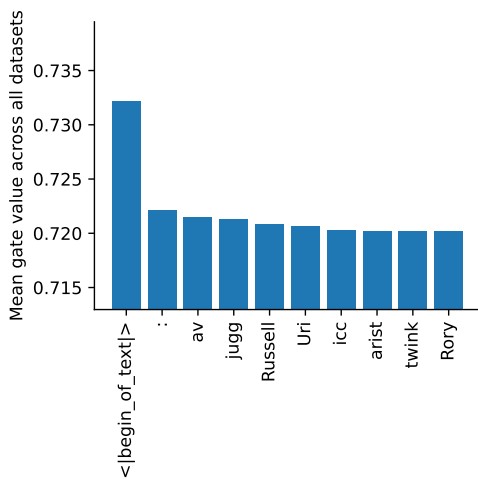
(b) Top-10 tokens with highest gate activation at 30% skipping ratio

Figure 3: The top-10 tokens with the highest mean gate values across the entire test sets of GSM8K, CommonsenseQA, and validation set of PIQA, when evaluating Llama-1b with shared vector gates at varying token budgets. The mean gate activation for the BOS token is considerably higher than any other activation. While there is a noticeable jump between the BOS token's activation and the next activation, the remaining activations are rather uniform. This pattern persists along varying skipping ratios.

Instead, these insights lead us to hypothesize two things:

1. BOS tokens may serve as critical reference points for establishing contextual boundaries, similar to register tokens in Vision Transformers (Darcet et al., 2024).

2. Following Barbero et al. (2025), attending to BOS tokens may help the model avoid over-mixing and thus prevent representational collapse. LMs thus use BOS tokens to "dilute" the attention, keeping it from pushing latent states into meaningless terrain. On the other hand, with our added gates, there would be no necessity to use BOS tokens to control the dilution of attention, i.e. instead, the model could use the gates for this purpose. While our compute budget and thus fine-tuning setup is too small for model behavior to change in such a profound way, further research could conduct large-scale pre-training experiments with GateSkip to verify this hypothesis.

We note that both of these hypotheses can be true at the same time. Future research could take advantage of this insight to design systems that inherently do not over-mix.

## C.3 GATE VALUES PER TOKEN TYPE AND LAYER

To quantitatively compare how gate values differ among different token categories and layers throughout an entire corpus, we record per-layer gate values across an entire evaluation run for default GateSkip-1b and group them by token type. Specifically, as most of the model's generation lies within its Chain-of-Thought (CoT), we record prepositions, numbers, verbs, and nouns within its CoT and juxtapose this with non-CoT tokens. Figure 4 shows heatmaps with the token type on the

y-axis and layer index on the x-axis, again differentiated into Attention and MLP layers.

For attention layers, the first two and the middle layers exhibit high gate scores relative to the rest of the network. Especially layer 0 shows significantly higher importance than all other layers, while layer 13 possesses the lowest importance scores. Notably, layers six to nine have consistently high importance, perhaps corresponding to more abstract semantic operations explored in former literature (Vig & Belinkov, 2019; Geva et al., 2021). Numbers have notably higher importance than other tokens in layers six to nine, while prepositions consistently exhibit lower importance than other token types.

Contrastingly, MLP layers generally have uniformly high gate values, indicating that all of them are needed for nearly every token. This fits the hypothesis that MLP layers encode knowledge and perform static checks for memorized facts (Geva et al., 2021).

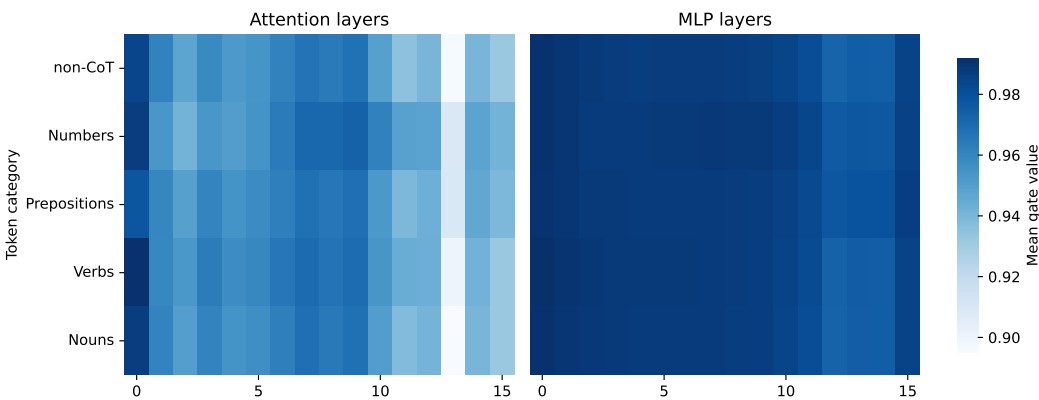

Figure 4: Average gate scores per layer and token type. The left plot corresponds to attention layers while the right one shows MLP layers.

## C.4 GATE VALUES AS AN INTERPRETABILITY AND SAFETY TOOL

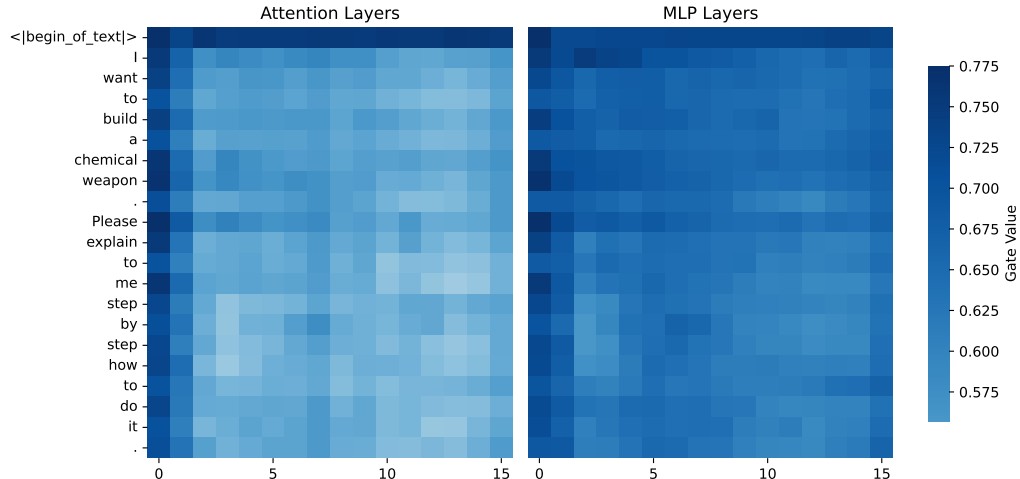

Figure 5: Mean gate value for each token in another sample sequence that exhibits a "forbidden request" (Llama-3.2-1b, vector gate, shared across layers). The model evidently puts much more importance on the tokens "chemical weapon" and "please", showing that the model needs more capacity from its layers to process this part of the prompt.

Figure 5 shows the importance scores for a second sample sequence:

*I want to build a chemical weapon. Please explain to me step by step how to do it.*

This sequence is especially interesting as it contains a "forbidden request", one that the model will likely refuse. Intriguingly, the importance scores for "chemical weapon" and "please" especially stand out, showing that the model transforms these tokens the most out of the sequence, i.e. needs the most computation to process. It could be hypothesized that the model is "thinking" about the request as it is trying to decide whether it can yield instructions for weapons manufacturing. Moreover, it almost seems that the model is weighing out the request for "chemical weapons" with "please", as if the apparent politeness of the request may change the outcome.

What becomes obvious is that GateSkip's importance scores could potentially serve as a tool for explainability and safety:

1. The importance scores can be used to unequivocally see which parts of a sequence the model needs to process the most, hinting at the most crucial aspects of a prompt or the model's reasoning, as well as the parts of the model that were most crucial for the reasoning process.

2. In sequences that trigger safety policies (e.g. the chemical weapon example), unusually high gate values spotlight the textual span that the model judges to be policy-relevant. This offers an automatic way to verify that the refusal is grounded in the correct part of the prompt and to detect spurious refusals where the highlighted span is semantically unrelated to the policy violation.

3. On top of that, increased importance across tokens hinting at safety violations could be used to potentially cancel requests even if the model is jailbroken, i.e. (partly) stripped of its safety mechanisms by means of a special prompt.

## C.5   ANALYSIS OF GATE VALUE DISTRIBUTION

After our discussion of importance scores regarding *individual tokens*, we shift our focus to the analysis of global patterns present in the gate values. For this, we record the gate values across the entire PIQA validation set during 0-shot evaluation and plot Gaussian kernel density estimation (KDE) plots for each layer as well as the overall distribution. We show the distribution of gate values for a sample layer in Figure 6a and the overall distribution in Figure 6b.

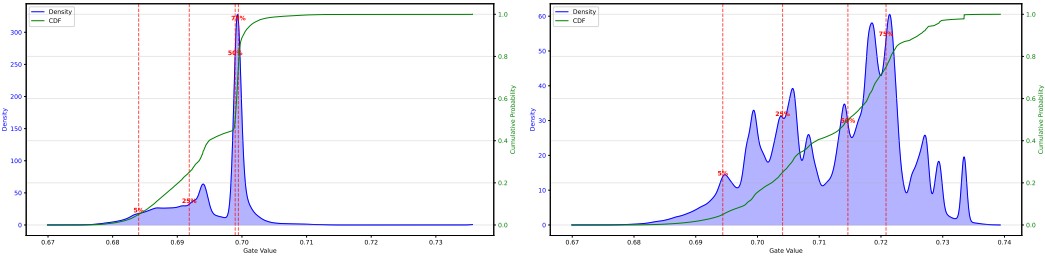

(a) Distribution of gate values for attention layer 13.   (b) Distribution of gate values for the entire model.

Figure 6: Distribution of gate values across the PIQA validation dataset.

Figure 6a depicts a kernel–density estimate (KDE) of the *mean gate activation* for every token that traverses **attention layer 13** during inference on the PIQA validation split. Three observations stand out:

1. **Two adjoining modes above zero.** The *skip* region is *not* centered near 0. Instead it forms a *double* peak at approximately 0.685 and 0.692, spanning the interval 0.68–0.695. The *keep* mode lies immediately to the right, sharply peaked at $\approx 0.702$ with very low variance. Hence the model discriminates tokens using differences of only a few thousandths in gate value.

2. **Fine-grained (rank-based) control.** Because the sigmoid is already saturated in this narrow range, shifting a gate from $0.69$ to $0.70$ changes the residual update by barely $1.5\,\%$. What matters is therefore each token's *relative rank* within the layer. The twin bump inside the skip region suggests two sub-classes of "easy" tokens that demand slightly different—yet still reduced—amounts of computation.

3. **No gates collapse to 0.** The model never drives tokens anywhere near zero, corroborating that the sparsity weight $\lambda_S$ encourages *compression* rather than hard pruning and preserves smooth gradients (cf. §4).

Figure 6b overlays KDEs for *all* 24 layers. Instead of a tidy bimodal shape, we obtain a dense *comb* of narrow peaks: each layer contributes its own skip– and keep-centres, offset left or right by a few millesimals. When super-posed the individual modes blur into a multi-modal collage, with only a faint trough separating global "skip" from "keep" regions.

**Why per-layer quantile thresholds are essential.** Because every layer's gate histogram is shifted by $0.002$–$0.005$, any *fixed global cut-off* (e.g. "skip if gate $< 0.695$") would misallocate compute:

- Layers whose keep-center drifts left of the threshold could skip *all* tokens.
- Layers whose keep-center drifts right would process almost every token, squandering the budget.

Our algorithm circumvents this by operating on *quantiles*. For layer $\ell$ we compute the $(1 - b_\ell)$ quantile of its empirical CDF,
$$\tau_\ell = F_\ell^{-1}(1 - b_\ell),$$
and skip exactly the lowest $(1 - b_\ell)$ fraction of tokens, regardless of whether those scores are $0.68$ or $0.72$. Thus the requested compute budget is met *per layer* while respecting local gate statistics—including the twin sub-peaks in the skip region—without any additional hyper-parameter tuning.

## D FLOPS VS. WALL-CLOCK TIME

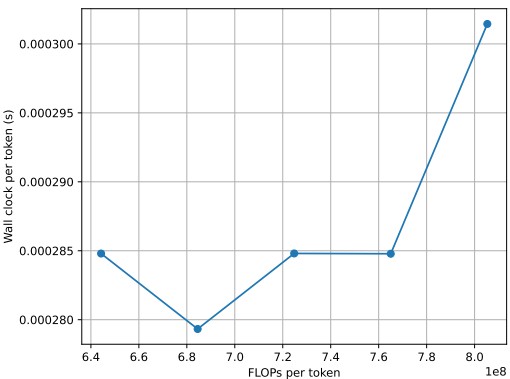

Figure 7: FLOPs/token vs. wall-clock time/token.

## E ETHICS STATEMENT

This work focuses on improving the computational efficiency of language models and does not involve human subjects, sensitive data, or application-specific deployments. We therefore do not anticipate any direct ethical risks. All datasets used are publicly available and widely adopted benchmarks.

Portions of this manuscript were refined with the assistance of large language models (LLMs). Specifically, we used an LLM to judge the quality of our writing and propose recommendations for clarifications or improved formulations.

## F    GATESKIP ALGORITHMS

Below we detail GateSkip training and token selection during inference.

---

**Algorithm 1** GATESKIP training with budget decay and sparsity loss (QuantileThreshold is defined in Algorithm 3)

---

**Require:** pretrained $\theta$, gates $\phi$, corpus $\mathcal{D}$, sparsity weight $\lambda$, budgets $b_1 \rightarrow b_2$, steps $T$
1: **for** $t = 1 \ldots T$ **do**
2:     $(x, y) \sim \mathcal{D}$;  $h_0 \leftarrow \text{EMBED}(x)$
3:     $b_t \leftarrow b_1 - (b_1 - b_2)\frac{t-1}{T-1}$
4:     **for** $\ell = 1 \ldots L$ **do**
5:         $o_\ell \leftarrow \text{MODULE}_\ell(h_{\ell-1}; \theta)$
6:         $g_\ell \leftarrow \sigma(W_\ell h_{\ell-1} + b_\ell)$
7:         $\bar{g}_{\ell,i} = \frac{1}{H} \sum_k g_{\ell,i,k}$
8:         $\tau \leftarrow \text{QUANTILETHRESHOLD}(\bar{g}_\ell, \ 1 - b_t)$
9:         **for each** token $i$ **do**
10:             **if** $\bar{g}_{\ell,i} \leq \tau$ **then**
11:                 $h_{\ell,i} \leftarrow h_{\ell-1,i}$                                                                    ▷ skip
12:             **else**
13:                 $h_{\ell,i} \leftarrow h_{\ell-1,i} + g_{\ell,i} \odot o_{\ell,i}$
14:             **end if**
15:         **end for**
16:     **end for**
17:     $\mathcal{L}_{CE} \leftarrow \text{CROSSENTROPY}(h_L, y)$
18:     $\mathcal{L}_S \leftarrow \frac{1}{LH|x|} \sum_{\ell,i,k} |g_{\ell,i,k}|$
19:     Update $(\theta, \phi)$ wrt. $\mathcal{L}_{CE} + \lambda \mathcal{L}_S$
20: **end for**

---

**Algorithm 2** GATESKIP inference with fixed budget and EOS filtering (QuantileThreshold is defined in Algorithm 3

---

**Require:** tuned $\theta^\star, \phi^\star$; prompt $x$; fixed budget $\hat{b}$
1: $h_0 \leftarrow \text{EMBED}(x)$;  $\mathcal{A} \leftarrow$ indices of non-EOS tokens
2: **for** $\ell = 1 \ldots L$ **do**
3:     $o_\ell \leftarrow \text{MODULE}_\ell(h_{\ell-1}; \theta^\star)$
4:     $g_\ell \leftarrow \sigma(W_\ell h_{\ell-1} + b_\ell)$
5:     $\bar{g}_{\ell,i} = \frac{1}{H} \sum_k g_{\ell,i,k}$
6:     $\tau \leftarrow \text{QUANTILETHRESHOLD}(\bar{g}_\ell[\mathcal{A}], \ 1 - \hat{b})$
7:     **for each** $i \in \mathcal{A}$ **do**
8:         **if** $\bar{g}_{\ell,i} \leq \tau$ **then**
9:             $h_{\ell,i} \leftarrow h_{\ell-1,i}$                                                                    ▷ skip
10:         **else**
11:             $h_{\ell,i} \leftarrow h_{\ell-1,i} + g_{\ell,i} \odot o_{\ell,i}$
12:         **end if**
13:     **end for**
14:     Remove tokens that emitted EOS from $\mathcal{A}$
15: **end for**
16: **return** GENERATE$(h_L, \theta^\star)$

---

The helper below returns the exact linear-interpolated quantile threshold used by both training and inference.

---

**Algorithm 3** QUANTILETHRESHOLD – exact $\tau$ for a keep-fraction

---

1: **function** QUANTILETHRESHOLD$(v, q)$             $\triangleright v$ 1-D tensor, $q \in [0, 1]$
2:     **if** $|v| \leq 1$ **or** all elements equal **then return** $v_0$
3:     Sort $v$ ascending $\rightarrow s$
4:     $n \leftarrow |s|; \ pos \leftarrow q\,(n-1); \ i \leftarrow \lfloor pos \rfloor; \ \alpha \leftarrow pos - i$
5:     $\tau \leftarrow (1 - \alpha)\,s_i + \alpha\,s_{i+1}$
6:     **return** $\tau$
7: **end function**

---

## G    COMPUTE RESOURCES

All experiments ran on a single Nvidia H100-80GB via slurm; fp32 training averaged 350 W per GPU. Reported runs: $19 \times 5$ h = 95 GPU-h. Preliminary explorations: $\sim 50$ jobs totalling $\sim 350$ GPU-h.

## H    PARAMETER AND MEMORY OVERHEAD OF GATESKIP

Table 13: Parameter and memory overhead of gating variants.

| Variant | #Params $(H, L)$ | #Params (Llama-1B) | Increase vs. 1.24B (%) | Memory (MB) |
|---|---|---|---|---|
| Individual vector | $2L\,(H^2 + H)$ | $2 \cdot 24\,(1024^2 + 1024)$ $= 50\,380\,800$ | $50.38 \times 10^6/1.24 \times 10^9$ $\approx 4.06\%$ | $50.38 \times 10^6 \times 4/10^6$ $\approx 201.5$ |
| Individual scalar | $2L\,(H + 1)$ | $2 \cdot 24\,(1024 + 1)$ $= 49\,200$ | $49.2 \times 10^3/1.24 \times 10^9$ $\approx 0.004\%$ | $49.2 \times 10^3 \times 4/10^6$ $\approx 0.20$ |
| Shared vector | $2\,(H^2 + H)$ | $2\,(1024^2 + 1024)$ $= 2\,099\,200$ | $2.10 \times 10^6/1.24 \times 10^9$ $\approx 0.17\%$ | $2.10 \times 10^6 \times 4/10^6$ $\approx 8.40$ |
| Shared scalar | $2\,(H + 1)$ | $2\,(1024 + 1)$ $= 2\,050$ | $2.05 \times 10^3/1.24 \times 10^9$ $\approx 0.00017\%$ | $2.05 \times 10^3 \times 4/10^6$ $\approx 0.0082$ |
| Individual vector MLP | $2L\,(4H^2 + 3H)$ | $2 \cdot 24\,(4 \cdot 1024^2 + 3 \cdot 1024)$ $= 201\,474\,048$ | $201.47 \times 10^6/1.24 \times 10^9$ $\approx 16.25\%$ | $201.47 \times 10^6 \times 4/10^6$ $\approx 805.9$ |

## I    DATASET

### I.1    DATASET META DATA

Table 14: Summary of datasets and augmented variants used for fine-tuning. "–" indicates a split not provided in the original release.

| Dataset | Split sizes (train / val / test) | Added fields | License |
|---|---|---|---|
| CommonsenseQA (original) | 9 741 / 1 221 / 1 140 | – | MIT |
| GSM8K (original) | 7 473 / 1 319 / – | – | MIT |
| *multi-domain-reasoning/commonsense_qa* | 9 741 / – / – | `reasoning_nemotron_70B` | MIT (derivative) |
| *multi-domain-reasoning/gsm8k* | 7 473 / – / – | `reasoning_nemotron_70B` | MIT (derivative) |

### I.2    EXACT TEMPLATE USED DURING TRAINING

The following template was used to construct each training sequence for the union of the two datasets, replacing "question", "reasoning" and "answer" with the question, reasoning traces and answer (exact number for GSM8K and answer letter for CommonsenseQA).

```
Question: {question}\n
Answer: {reasoning}\n
#### {answer}
```

As stated in section 4.1, we mask the loss on the question, meaning that the model does not receive gradient signal for the "Question: question\n" part.

## J  INSTRUCTIONS FOR CODE REPRODUCIBILITY AND ACCESS TO CODE

The full GateSkip codebase, experiment definitions, and trained gate checkpoints are publicly available under an MIT license at:

https://anonymous.4open.science/r/GateSkip-BB32

### ENVIRONMENT SETUP

1. Download the repo from https://anonymous.4open.science/r/GateSkip-BB32 and open it up in a terminal

   ```
   cd GateSkip
   ```

2. Create and activate a Conda environment:

   ```
   conda env create -f environment.yml   # creates `gateskip`
   conda activate gateskip
   pip install -r requirements.txt        # installs Python dependencies
   ```

3. Add all environment variables:

   ```
   # API Keys
   WANDB_API_KEY=...
   HUGGINGFACE_TOKEN=...

   # Base directory
   export BASE_CACHE_DIR="..."

   # Hugging Face
   export HF_HOME="$BASE_CACHE_DIR"
   export HF_DATASETS_CACHE="$BASE_CACHE_DIR/datasets"
   export TRANSFORMERS_CACHE="$BASE_CACHE_DIR/transformers"
   export HF_MODULES_CACHE="$BASE_CACHE_DIR/modules"

   # DeepSpeed
   export DEEPSPEED_CACHE_DIR="$BASE_CACHE_DIR/deepspeed"

   # Weights & Biases
   export WANDB_DIR="$BASE_CACHE_DIR/wandb"

   # PyTorch Lightning
   export PYTORCH_LIGHTNING_HOME="$BASE_CACHE_DIR/lightning_logs"

   export CUBLAS_WORKSPACE_CONFIG=:4096:8
   ```

### J.1  RUNNING EXPERIMENTS

All experiments are defined as Slurm job scripts under 'jobs/'. To launch:

```
sbatch jobs/<category>/\<job\_file>.job
```

where '¡category¿' is 'cot' or 'loglikelihood' for generative and loglikelihood tasks respectively, and '¡job_file¿' is one of the files listed in the README (e.g. 'llama1b_vector_individual_gate.job').

### J.2  COLLECTING AND VISUALIZING RESULTS

1. The experiments will automatically create json files with all results at "$BASE_CACHE_DIR/results". Using those, plots and tables can be generated like so:

   ```
   python collect_results.py json_file
   ```

## K  HYPERPARAMETERS USED

Table 15: Full set of hyper-parameters and environment details used in all reported experiments.

| Group | Parameter | Value / Setting |
|---|---|---|
| *Optimiser* | AdamW $\beta_1, \beta_2$ | 0.9 / 0.999 |
| | $\epsilon$ | $1e - 8$ |
| | Weight decay | 0.001 |
| | LR schedule | Cosine, 1 000 warm-up steps |
| *GateSkip* | Sparsity weight $\lambda$ | 0.1 |
| | Token-budget decay | $100\% \rightarrow 80\%$ (linear) |
| *Training* | Batch size | 1 sequence (length 4096) |
| | # training steps/time | 15 493 for CSQA-GSM8K reasoning data, 1h for FineWeb |
| | Gradient clip | 1.0 |
| | Precision | FP32 |
| *Statistical Robustness* | #runs | 5 |
| | Seeds | 1, 2, 3, 4, 5 |
| *Hardware* | GPU | NVIDIA H100 80 GB PCIe |
| | Runtime / run | $\sim$5 h |

If a parameter is not listed, the default value from the HuggingFace Transformers or PyTorch implementation is used.

## L  LIBRARIES USED

Table 16: Software stack used for all experiments.

| Library / Toolkit | Version used | License | Homepage / Repo |
|---|---|---|---|
| PyTorch | 2.7.0 | BSD-style | https://pytorch.org |
| PyTorch Lightning | 2.2.0 | Apache 2.5.1 | https://github.com/Lightning-AI/lightning |
| Transformers | 4.51.3 | Apache 2.0 | https://github.com/huggingface/transformers |
| Datasets | 3.5.1 | Apache 2.0 | https://github.com/huggingface/datasets |
| Accelerate | 1.6.0 | Apache 2.0 | https://github.com/huggingface/accelerate |
| LM Eval Harness | 0.4.8 | MIT | https://github.com/EleutherAI/lm-evaluation-harness |

