# OpenReview forum: "What Layers When: Learning to Skip Compute in LLMs with Residual Gates"
_ICLR.cc/2026/Conference — ICLR 2026 Poster_

### Official Review · Reviewer_ocRk · 2025-10-27

**Soundness:** 3
**Presentation:** 3
**Contribution:** 3
**Rating:** 8
**Confidence:** 3

**Summary:**

The paper proposes GateSkip, a lightweight, differentiable gating mechanism added to the residual stream of decoder-only LMs.
The gate module is a linear+sigmoid function that multiplies the output after each attention and MLP branch. Gates are trained with an L2 penalty on top of a pretrained model. At inference, token-wise gate scores are converted to per-layer quantile thresholds to keep a target budget of tokens and skip the rest. Extensive experiments on various benchmarks, previous methods, model sizes, and ablations are conducted.

**Strengths:**

- The paper is well written. The introduction section clearly states the research problem, issues in current methods, and the contributions.
- The method is simple and effective.
- The experiments are solid. I particularly like the ablation results in Table 5.

**Weaknesses:**

- The viewpoint “early layers allocate more computation to the beginning-of-sequence token and punctuation, while deeper layers become increasingly selective and focus on content-bearing words” may be overclaimed based on Figures 2 and 4.
- The overhead of HxH gates per branch, including both parameters and computation, may not be negligible.

**Questions:**

- Did you try non-uniform per-layer budgets (b_l), such as allocating more compute at mid-depth?
- There are many interesting results in the experiments. Could you offer some explanations for these results?
	- The difference between generative and log-likelihood benchmarks: previous methods show large degradation on generative benchmarks, while even random skipping is not too bad on log-likelihood benchmarks (Tables 1–3).
	- GateSkip achieves better performance as saved compute increases in the PIQA-Gen task (Table 4).
- If I skip some tokens at the first layer, it seems that I have no KV cache to copy from. How do you handle this issue?

---

> ### Author Response · Authors · 2025-11-21
>
> We thank the reviewer for their thoughtful advice and detail clarifications and additions below:
>
> **Weaknesses:**
> 1. We have weakened our claims in Appendix C, as upon reflection we agree that while the Figures give a good indication of this behavior further research is needed to manifest these claims. We will soon update the pdf with these changes as well as further extensive polishing.
> 2. In practice, the overhead of HxH gates is very small relative to the transformer blocks they extend. For example, in Llama-3.2-1B, a full linear gate adds 0.004% additional parameters in the scalar-gate variant and ~4% in the vector-gate variant, while contributing <0.4% of total Flops per forward pass (measured at 1024 tokens). During inference, gates are computed only once per layer per token and do not require storing additional activations, so peak memory usage is unchanged too. As shown in Sec. 4 and especially our throughput experiments in Sec. 4.6, the theoretical FLOP overhead is far outweighed by the 15-50% compute savings gained through skipping tokens. Table 13 in App. G shows a detailed breakdown of gate overhead.
>
> **Questions:**
> 1. Real-world gains of our method rely on increasing batch size for smaller budgets, so maximum possible batch size depends on the highest allocated budget across the model, meaning that lower budgets at the beginning and end would only make the GPU more idle. Our analysis on gate values in Appendix C further indicates that early layers need more compute than middle and end layers.
> 2. Result explanations:
> - Gen. vs. log.-lik. results: benchmarks like PIQA or CSQA have 2-4 answer options, so for log-lik eval the worst possible result is chance level at 25% for CSQA and 50% for PIQA. For their generative variants, an answer is only counted if the model correctly outputs \<answer\>X\<\/answer\> so much lower accuracy is possible. This is why accuracy seems higher in the log-lik setting.
> - Increasing performance with larger savings: our hypothesis is that because GateSkip is trained to filter out unneeded computation, it hence reduces noise so that the important information is much better preserved, thereby increasing performance.
> 3. When skipping a layer, GateSkip upwards copies the KV-cache item from the layer below, analogous to other skipping/early-exiting methods.
>
> Again, we are very grateful for the nice and helpful feedback. If our additions clarified the reviewer’s concerns and questions, we would kindly ask the reviewer to increase their rating and confidence.

---

> > ### Author Response · Authors · 2025-11-25
> >
> > > UPDATE: We added a new analysis section comparing gate activations across different token types and layers (Appendix C.3). While we cannot show the figure in this comment, we invite you to have a look at the revised pdf, otherwise the text is shown below. We think that these findings give additional evidence for the hypothesis that “early layers allocate more computation to the beginning-of-sequence token and punctuation, while deeper layers become increasingly selective and focus on content-bearing words” - we will nonetheless weaken the claim.
> >
> >
> > ### GATE VALUES PER TOKEN TYPE AND LAYER
> >
> > To quantitatively compare how gate values differ among different token categories and layers
> > throughout an entire corpus, we record per-layer gate values across an entire evaluation run for
> > default GateSkip-1b and group them by token type. Specifically, as most of the model’s generation
> > lies within its Chain-of-Thought (CoT), we record prepositions, numbers, verbs, and nouns within its
> > CoT and juxtapose this with non-CoT tokens. Figure 4 shows heatmaps with the token type on the y-axis and layer index on the x-axis, again differentiated into Attention and MLP layers.
> >
> > For attention layers, the first two and the middle layers exhibit high gate scores relative to the rest of
> > the network. Especially layer 0 shows significantly higher importance than all other layers, while
> > layer 13 possesses the lowest importance scores. Notably, layers six to nine have consistently high
> > importance, perhaps corresponding to more abstract semantic operations explored in former literature
> > (Vig & Belinkov, 2019; Geva et al., 2021). Numbers have notably higher importance than other
> > tokens in layers six to nine, while prepositions consistently exhibit lower importance than other token
> > types.
> > Contrastingly, MLP layers generally have uniformly high gate values, indicating that all of them
> > are needed for nearly every token. This fits the hypothesis that MLP layers encode knowledge and
> > perform static checks for memorized facts (Geva et al., 2021).

---

### Official Review · Reviewer_JNXt · 2025-10-31

**Soundness:** 3
**Presentation:** 2
**Contribution:** 3
**Rating:** 6
**Confidence:** 3

**Summary:**

GateSkip is a new residual-stream gating method for decoder-only LLMs that enables token-wise layer skipping during inference. It uses smooth, differentiable gates fine-tuned on pretrained models to decide token importance, avoiding the instability of earlier adaptive compute methods. Tests on Llama and Gemma show 15–20% compute savings with over 90% accuracy retention and compatibility with techniques like quantization and pruning. Gate analysis also reveals higher importance for BOS tokens and punctuation.

**Strengths:**

1. The proposed method achieves a strong compute–accuracy trade-off, reducing computation by up to 15% while maintaining over 90% accuracy.
2. It is compatible with other efficiency techniques, further enhancing its practical value.
3. The authors conducted comprehensive ablation studies to validate their design choices.

**Weaknesses:**

1. It seems the method is sensitive to architecture choice. The placement of the gate is important. I'd prefer more discussion on why this is the case.
2. Similarly, this method also is sensitive to the choice of the gate. I wonder, besides MLP, whether the authors have tried other projection methods.


Presentation:
1. I am not sure why there is an empty line after two paragraphs in the Related Work part (e.g., Line 95~97, 104~105).
2. According to ICLR format for tables, table captions should come before the table.
3. Line 472: the reference to the appendix broke.

**Questions:**

1. What is the intuition behind this design?
2. In the Related Work part, the authors mention that layer pruning, token pruning, etc., operate on different axes of efficiency from your work, and thus those are orthogonal. Could you elaborate more on what specifically those axes are and why they are different from yours? It seems your method also helps for layer skipping in the sense of the attention layer or MLP layer.

---

> ### Author Response · Authors · 2025-11-21
>
> We thank the reviewer for their thoughtful advice and detail clarifications and additions below:
>
> **Weaknesses:**
>
> 1. We ablated shared gates (one shared gate for all layers) vs. separate ones (each layer has its own gate). The second variant performed much better, because it has (1) separate params, and (2) can learn different dynamics for each layer. So the placement of the gate is not directly important, it's rather that with higher capacity, the model achieves higher performance.
> 2. We ablated linear+sigmoid gate (i.e. linear patterns) vs. MLP (i.e. complex non-linear patterns) to answer "Does the gate need to be able to capture complex non-linear patterns?". The answer is no, the linear gate suffices. We also tried scalar gates, i.e. even less complex patterns (also in the ablations table), and they also worked fine. Other projection methods like diagonal/triangular gates etc. wouldn't add anything meaningful to this analysis in our opinion because they are somehow in between scalar-gate and MLP-gate on the spectrum of "gate complexity".
> 3. *Presentation:* we will upload a new pdf with all these fixes as well as extensively polished writing and the additional explanations from this rebuttal in the following days.
>
>
> **Questions:**
>
> 1. *Intuition behind our design:*
> - LSTMs have gates that control how much information should be kept at each timestep. Transformers just blindly add any output onto their residual stream. Hence, it makes sense to add a gate here too to more carefully choose what information from Attn/MLP modules to incorporate into the residual stream.
> - Different Transformer layers specialize in different operations (Geva et al., 2020). So if some operation isn't needed for a given token/task, we can learn a mechanism that chooses to skip the operation.
>
> 2. By “different axes of efficiency,” we mean that existing methods reduce computation by permanently modifying the model, while GateSkip reduces computation dynamically per token during inference. Quantization reduces precision (bits per weight/activation), pruning removes parameters or entire layers (width/depth), and token/KV pruning reduces sequence length. In contrast, GateSkip adaptively skips layer computation only for low-importance tokens, without changing model capacity or structure. These approaches hence optimize orthogonal dimensions and can be combined. Moreover, GateSkip operates on a similar axis as early exiting methods.
>
> We again thank the reviewer for the thoughtful feedback. If our additions clarified the reviewer’s concerns and questions, we kindly ask the reviewer to increase their rating and confidence.
>
> **References:**
> Geva et al., "Transformer Feed-Forward Layers Are Key-Value Memories", 2020

---

> > ### Comment · Reviewer_JNXt · 2025-11-24
> >
> > Thank you for clarifying my concerns. I will keep my score and raise the confidence since it already reflects my assessment.

---

### Official Review · Reviewer_5ZF6 · 2025-10-31

**Soundness:** 3
**Presentation:** 4
**Contribution:** 3
**Rating:** 4
**Confidence:** 3

**Summary:**

The paper introduces GateSkip, a residual-stream gating mechanism that enables token-wise layer skipping in decoder-only LMs. Each Attention/MLP branch outputs is multiplied by a sigmoid(linear) gate; gates are trained jointly with the backbone using LM loss + sparsity penalty, and at inference the per-token gate scores are quantile-thresholded to enforce a layer-wise compute budget (skipped tokens copy hidden states and KV cache upward).

On Llama/Gemma models, GateSkip claims up to 15–20% compute savings while retaining >90% of baseline accuracy on long-form reasoning, and near-50% savings with matched accuracy on instruction-tuned models; it also composes with quantization, pruning, and self-speculative decoding. Methods, hyperparameters, and seeds are documented (with single-seed runs and bootstrap variance).

**Strengths:**

### Originality

* Reframes adaptive depth via residual gating (no discrete router during training), then converts scores to hard skips at test time; per-token budgets encourage fine-grained allocation.

### Quality

* Clear mechanism (Eq. 2), objective (Eq. 3–5), token selection (Eq. 6), init, and overhead; ablations cover gate shape, sharing, placement, and skipping granularity.
* Side-by-side comparisons against MoD, CALM, FREE, and static baselines under identical protocols.

### Clarity

* Figures/tables report compute budgets and (stated) seed counts; the experimental setup is centralized in §4.1 with dataset/model details.

### Significance

* Achieves compute–accuracy trade-offs on generative tasks where prior methods degrade; composes with 4-bit quantization and self-speculative decoding.

**Weaknesses:**

1. Most results are single-seed with bootstrap CIs; for stability claims this is thin. Please add ≥5 seeds for key curves and show error bars.

2. Backbone fine-tuning confounds “accuracy gains at 0%.” The setup jointly fine-tunes the backbone while training gates, so reported accuracy lifts at 0% compute savings likely reflect backbone adaptation.

3. End-to-end speed is modest/non-monotonic. The latency/throughput table shows limited and non-monotonic gains vs. skip rate; please report FLOPs vs. wall-clock and break out gate overhead. A kernel-optimized implementation (vLLM/custom CUDA) would also strengthen real-world evidence.

4. Positioning vs. strongest baselines. MoD is included, but add a router-tuned MoD on the same frozen backbone and compare stability (loss curves, LR sensitivity) to substantiate the “stable training” claim.

**Questions:**

1. Isolating gate effect. Please report frozen-backbone results and an equally fine-tuned baseline (no gates) to separate backbone tuning gains from gating.

2. Seed count / CIs. For the main curves (Tables 1–3), can you re-run with ≥5 seeds and include 95% CIs?

3. Wall-clock accounting. What is the per-layer gate FLOPs/latency? Please add a column breaking down FLOPs saved vs. FLOPs added and relate this to Table 10’s latency/throughput.

4. Router baseline stability. Under your setup, how does router-tuned MoD behave (loss curves / collapse rate) vs. GateSkip? A stability plot would support the abstract’s claim.

5. Gating loss + budgets. Why L2 on gate activations (Eq. 4) vs L1/KL? Sensitivity to λS and to the budget decay schedule?

6. Where does it help most? Any breakdown of skips by layer/token type on reasoning tasks (build on your BOS/punctuation analysis) to guide deployments?

---

> ### Author Response · Authors · 2025-11-21
>
> We thank the reviewer for their extensive and helpful feedback. We summarize improvements and clarifications below:
>
>
> 1. **Isolate gate effect:** To keep comparisons fair, we always adapt the backbone, even for the random baseline. We are currently running experiments for frozen backbone with a linear gate, and one with an MLP gate. We will update the results with the final numbers within a few days.
>
> 2. **Different seeds:** We have added 5-seed results for our baseline, GateSkip-1b as well as several other experiments, evaluating on GSM8k, CommonsenseQA, PIQA and MMLU (partial results with mean and standard deviation shown below). We are trying our best to update all results as soon as possible, but the experiments take time, and we will update all numbers within a few days. We further aim to re-run all experiments of our paper with 5-seeds before the end of the rebuttal period.
>
> Table 1: results representing the average of CSQA, GSM8K, MMLU and PIQA (note: before, it was only CSQA and GSM8K, to strengthen our evaluation we added the other two metrics here as well).
> | Method                     | Gen @0%        | Gen @5%        | Gen @10%       | Gen @15%       | Gen @20%       | Gen @25%       | Gen @30%       | Gen @45%       | Gen @60%       | LL @0%        | LL @15%       | LL @30%       | LL @45%       | LL @60%       |
> |----------------------------|----------------|----------------|----------------|----------------|----------------|----------------|----------------|----------------|----------------|---------------|---------------|---------------|---------------|---------------|
> | Random skipping baseline   | 31.8 ± 1.8     | 13.4 ± 1.9     | 3.6 ± 0.6      | 1.6 ± 0.3      | 1.2 ± 0.2      | 0.8 ± 0.2      | 0.5 ± 0.1      | 0.0 ± 0.0      | 0.0 ± 0.0      | 49.1 ± 0.1    | 25.6 ± 0.3    | 23.6 ± 0.3    | 23.4 ± 0.2    | 23.7 ± 0.1    |
> | GateSkip (separate vector) | 20.6 ± 2.0     | 21.3 ± 1.6     | 22.0 ± 1.9     | 22.9 ± 2.4     | 23.9 ± 1.8     | 22.9 ± 1.6     | 21.3 ± 1.5     | 5.1 ± 1.4      | 2.7 ± 0.9      | 47.4 ± 0.2    | 38.9 ± 0.5    | 31.7 ± 2.0    | 27.9 ± 1.1    | 26.5 ± 0.6    |
> | MoD                        | 17.8 ± 1.0     | 10.7 ± 1.2     | 7.7 ± 0.8      | 6.7 ± 1.1      | 6.7 ± 1.1      | 5.6 ± 0.7      | 4.8 ± 1.2      | 0.9 ± 0.5      | 0.4 ± 0.2      | 44.2 ± 0.7    | 31.9 ± 1.5    | 29.3 ± 2.4    | 26.7 ± 0.4    | 26.0 ± 0.6    |
> | MoD (router tuned)         | 0.0 ± 0.0      | 0.0 ± 0.0      | 0.0 ± 0.0      | 0.0 ± 0.0      | 0.0 ± 0.0      | 0.0 ± 0.0      | 0.0 ± 0.0      | 0.0 ± 0.0      | 0.0 ± 0.0      | 51.3 ± 0.1    | 23.7 ± 0.0    | 23.0 ± 0.2    | 22.9 ± 0.1    | 22.9 ± 0.0    |
> | GateSkip (L1 loss)         | 24.0 ± 1.6     | 23.9 ± 1.8     | 23.8 ± 2.1     | 23.5 ± 2.6     | 22.8 ± 3.5     | 21.8 ± 2.7     | 18.4 ± 1.3     | 5.1 ± 0.9      | 3.4 ± 1.6      | 47.5 ± 0.1    | 37.6 ± 0.3    | 29.2 ± 1.8    | 27.1 ± 2.2    | 26.3 ± 2.2    |
> | GateSkip (KL loss)         | 26.5 ± 1.0     | 16.9 ± 2.9     | 13.0 ± 1.2     | 11.9 ± 0.3     | 11.1 ± 0.7     | 10.1 ± 1.3     | 10.3 ± 1.5     | 8.6 ± 3.7      | 4.1 ± 3.0      | 45.6 ± 0.2    | 26.6 ± 0.2    | 24.5 ± 1.0    | 23.0 ± 0.2    | 23.1 ± 0.2    |
>
>
> 3. **Kernel-optimized implementation:** We implemented a Triton kernel for our model and are currently rerunning the throughput experiments on the vLLM engine. We will give an update with FLOPs vs. wall-clock time per token in the coming days.
>
> 4. **MoD-stability:** We are rerunning the MoD and router tuned MoD runs with 5 seeds. While the experiments take some time, we will provide results plus stability analysis in the coming days.
>
> 5. **Different gate losses:** We have added an ablation for gates trained with L2 vs L1 and KL-div. Losses (see table above). As you can see, normal GateSkip outperforms the two other options.
>
> 6. **Analysis of importance based on token type:** We are currently running an experiment recording average gate activations per token type (e.g. numbers vs. non-numbers, prepositions vs. nouns vs. verbs, CoT-tokens vs. non-CoT-tokens) as well as average activations per layer. We will add these results in the coming days as well.
>
> We will polish our paper extensively based on the final numbers as soon as possible and update the pdf as well as give an update here.
>
> We again thank the reviewer for their nice and insightful feedback. If these additions clarify the reviewer’s concerns and questions, we kindly ask the reviewer to increase their rating and confidence.

---

> > ### Author Response · Authors · 2025-11-22
> >
> > UPDATE: We updated the results in Table 1 and attach a second Table below with accuracies per benchmark at 15% compute savings.
> >
> > Table 2: accuracies for each individual benchmark at 15% compute savings
> > | Method                     | CSQA (Gen.)      | GSM8K (Gen.)      | MMLU (Gen.)       | PIQA (Gen.)       | MMLU-Stem         | HellaSwag         | CSQA             | PIQA             | OpenBookQA       | WinoGrande       |
> > |----------------------------|------------------|-------------------|-------------------|-------------------|-------------------|-------------------|------------------|------------------|------------------|------------------|
> > | Random skipping baseline   | 3.3 ± 0.5        | 0.1 ± 0.1         | 1.9 ± 0.7         | 1.1 ± 0.5         | 24.0 ± 0.3        | 30.9 ± 0.6        | 20.3 ± 1.1       | 53.7 ± 0.9       | 15.7 ± 0.7       | 51.1 ± 0.3       |
> > | MoD                        | 6.1 ± 3.1        | 1.8 ± 0.3         | 9.6 ± 2.9         | 9.4 ± 2.6         | 26.0 ± 0.7        | 38.2 ± 1.6        | 20.9 ± 0.9       | 59.9 ± 2.2       | 18.6 ± 1.4       | 52.2 ± 0.7       |
> > | MoD (router tuned)         | 0.0 ± 0.0        | 0.0 ± 0.0         | 0.0 ± 0.0         | 0.0 ± 0.0         | 23.1 ± 0.2        | 28.9 ± 0.4        | 19.9 ± 0.2       | 53.6 ± 0.4       | 13.8 ± 0.5       | 49.1 ± 1.4       |
> > | GateSkip (KL loss)         | 9.8 ± 1.1        | 3.2 ± 0.8         | 19.8 ± 3.0        | 14.6 ± 3.5        | 21.3 ± 0.0        | 30.0 ± 0.1        | 18.9 ± 0.1       | 56.3 ± 0.1       | 17.0 ± 0.6       | 50.1 ± 0.9       |
> > | GateSkip (L1 loss)         | 33.7 ± 0.3       | 9.1 ± 1.8         | 28.0 ± 7.5        | 23.3 ± 3.9        | 29.6 ± 0.7        | 39.8 ± 0.9        | 33.1 ± 2.5       | 69.9 ± 0.6       | 21.6 ± 0.8       | 51.8 ± 1.4       |
> > | GateSkip (separate vector) | **35.3 ± 1.8**   | **9.0 ± 1.3**     | **22.8 ± 7.8**    | **24.3 ± 6.1**    | **30.9 ± 1.2**    | **39.1 ± 1.3**    | **36.2 ± 1.2**   | **70.5 ± 0.6**   | **22.7 ± 0.4**   | **52.7 ± 0.8**   |

---

> > > ### Author Response · Authors · 2025-11-25
> > >
> > > > UPDATE: We added a new analysis section comparing gate activations across different token types and layers (Appendix C.3). While we cannot show the figure in this comment, we invite you to have a look at the revised pdf, otherwise the text is shown below.
> > >
> > >
> > > ### GATE VALUES PER TOKEN TYPE AND LAYER
> > >
> > > To quantitatively compare how gate values differ among different token categories and layers
> > > throughout an entire corpus, we record per-layer gate values across an entire evaluation run for
> > > default GateSkip-1b and group them by token type. Specifically, as most of the model’s generation
> > > lies within its Chain-of-Thought (CoT), we record prepositions, numbers, verbs, and nouns within its
> > > CoT and juxtapose this with non-CoT tokens. Figure 4 shows heatmaps with the token type on the y-axis and layer index on the x-axis, again differentiated into Attention and MLP layers.
> > >
> > > For attention layers, the first two and the middle layers exhibit high gate scores relative to the rest of
> > > the network. Especially layer 0 shows significantly higher importance than all other layers, while
> > > layer 13 possesses the lowest importance scores. Notably, layers six to nine have consistently high
> > > importance, perhaps corresponding to more abstract semantic operations explored in former literature
> > > (Vig & Belinkov, 2019; Geva et al., 2021). Numbers have notably higher importance than other
> > > tokens in layers six to nine, while prepositions consistently exhibit lower importance than other token
> > > types.
> > > Contrastingly, MLP layers generally have uniformly high gate values, indicating that all of them
> > > are needed for nearly every token. This fits the hypothesis that MLP layers encode knowledge and
> > > perform static checks for memorized facts (Geva et al., 2021).

---

### Official Review · Reviewer_vk8d · 2025-10-31

**Soundness:** 3
**Presentation:** 3
**Contribution:** 3
**Rating:** 6
**Confidence:** 4

**Summary:**

This paper introduces GateSkip, a method for enabling token-wise layer skipping in decoder-only transformer language models. GateSkip adds a small linear gate with sigmoid activation to each attention and MLP branch in the transformer. During training, the gates are optimized to remain sparse while preserving language modeling accuracy. At inference time, the gates produce token-level importance scores, allowing the model to skip the least important tokens in each layer. Skipped tokens have their hidden states and key-value cache entries copied upward.

GateSkip is evaluated on Llama and Gemma models and find that it reduces computation by up to 15% on reasoning tasks while retaining over 90% of the original accuracy. On instruction-tuned models, GateSkip improves accuracy even at full compute and matches baseline quality with around 50% compute savings. Analyzing the learned gate values reveals that early layers allocate more compute to the BOS token and punctuation, while deeper layers focus compute on content words.

**Strengths:**

* **Novel approach**: The paper introduces a novel approach to layer skipping, GateSkip, which is different from existing methods such as Mixture-of-Depths (MoD) and early-exit methods.

* **Smooth and differentiable gates**: The use of smooth and differentiable gates allows for stable training and fine-tuning on top of pre-trained models, which is a significant advantage over existing methods that rely on hard, discrete decisions.

* **Token-level control**: GateSkip provides fine-grained control at both the token and module level, enabling nuanced allocation of compute resources.

* **Compatibility with other efficiency techniques**: The paper shows that GateSkip can be combined seamlessly with other efficiency techniques such as quantization, pruning, and self-speculative decoding.

* **Strong experimental results**: The paper presents strong experimental results, demonstrating that GateSkip can reduce computation by up to 15% while retaining over 90% of the original accuracy on reasoning tasks.

* **Insights into transformer information flow**: The analysis of learned gate values provides interesting insights into the information flow within transformers, such as the allocation of compute resources to different tokens and layers.

**Weaknesses:**

* **Efficiency**:  Provide more context on the setting for the benchmarking. For the throughput reported numbers in line 447. Do you increase the batch size as you increase the number of tokens skipped ?  How does this method affect the peak memory utilization compared to the baseline when there is no skipping ? Is this for pre-fill or decode ? what is the total number of tokens etc.

* **Analysis**: For the most part I like the analysis in the paper. I would suggest to maybe emphasize some other things more e.g why do we have significantly better performance for this method when using it on instruction tunede models ? Am I right to tell from Figure 1 C) that the performance of GateSkip-3b-Instruct is almost 50% at 0% skipped while Llama-3b-Instruct is only about 35% ? This is a huge difference in quality. What accounts for this ?

* **Writing and Clarity**: The work could benefit from some more polishing and make the work more clear e.g on Line 472: Appendix ??

**Questions:**

See Weaknesses

---

> ### Author Response · Authors · 2025-11-21
>
> We thank the reviewer for their thoughtful and helpful advice. Clarifications are detailed below:
>
> **Efficiency:**
> - We report throughput while scaling the batch size to fill out GPU memory as skip ratio increases. This isolates the real deployment benefit of adaptive token skipping, where reduced computation per token directly translates into larger batch sizes and higher throughput.
> - Peak memory utilization does not increase compared to baseline, because skipped tokens reuse KV cache entries from the preceding layer and gating does not require storing additional activations.
> - This setting corresponds to autoregressive decode (token-by-token), not prefill.
> - We fix a batch size of 64 during generation, as well as 1024 generated tokens.
> - We use H100 80GB with BF16 inference.
> - We will add these details to the paper in the coming days.
>
>
> **Analysis:**
> - **Why instruct models are so much better**: We can only hypothesize, but perhaps this is because the instruction tuning gives a much better prior on the goal of a given task so that the model can better judge whether tokens are easy vs hard.
>
> **Writing and Clarity:**
> - We fixed the broken reference, and are actively polishing the work. We will extensively re-read and improve the writing within the rebuttal period.
>
> We are very grateful for the insightful feedback and suggestions. If these additions clarified the reviewer’s concerns and questions, we would kindly ask the reviewer to increase their rating and confidence.

---

> > ### Comment · Reviewer_vk8d · 2025-11-24
> >
> > Thank the authors for clarifying my concerns. I will keep my score, as I think it reflects my assessment of the work

---

### Author Response · Authors · 2025-12-01

We thank all reviewers for their thorough analysis of our work and their helpful feedback. We will  summarize the changes to our paper based on the reviews below:

1. We added 5-seed results for our main Tables 1 and 2, now reporting means and standard deviations for statistical robustness.
2. We added an ablation comparing (a) GateSkip with a frozen backbone to (b) default GateSkip and (c) random skipping, isolating the extent of the backbone adapting to the skipping task (Table 5).
3. We further added an ablation comparing different gate losses, (a) L1, (b) L2, and (c) KL-divergence on the gates (Table 5).
4. We added an additional SOTA baseline, "Mixture of Depths Router Tuning", to our main Tables 1 and 2.
5. We quantitatively analyzed gate scores across different token types (e.g. numbers, verbs, nouns, CoT vs. non-CoT tokens) and add an additional analysis section to our paper in Appendix C.
6. We developed a kernel-optimized vLLM implementation of our method and updated the real-world evaluation in Section 4.6 with additional results showing FLOPs vs. wall-clock time per token.
7. We  polished the writing of our work, adding more clarity to the entire paper and removing some minor render errors.

We believe that these additions address all the weaknesses and questions the reviewers raised. We again present our gratitude to the reviewers and the AC for their careful work.

---

### Meta-Review · Area_Chair_r2a5 · 2026-01-12

**Summary:**

This is a borderline paper exploring a different way of doing layer skipping, a topic that has been studied quite a bit. Reviewers find the proposed techniques improving over the current approaches. They asked for number of ablations and raised questions about efficiency of the proposed approach. Authors provided a thorough response and reviewers updated their scores suggesting acceptance.

**Reviewer Concerns:**

Reviewers asked for additional experiments ablating the propose method and raise questions about efficiency of the approach.

**Reviewer Scores:**

Most reviewers updated their scores during discussion. Reviewer 5ZF6 didnt update their score, but looking at authors response i think they would also have updated to 6.

---

### Decision · Program_Chairs · 2026-01-26

Accept (Poster)